EMBO
Molecular Medicine

# Proteasome subunit *PSMC3* variants cause neurosensory syndrome combining deafness and cataract due to proteotoxic stress

Ariane Kröll-Hermi[1,2,†], Frédéric Ebstein[3,†], Corinne Stoetzel[1,†], Véronique Geoffroy[1,†], Elise Schaefer[1,4], Sophie Scheidecker[1,5], Séverine Bär[6], Masanari Takamiya[2], Koichi Kawakami[7,8], Barbara A Zieba[3], Fouzia Studer[9], Valerie Pelletier[4,9], Carine Eyermann[10], Claude Speeg-Schatz[11], Vincent Laugel[1,12], Dan Lipsker[13], Florian Sandron[14] (iD), Steven McGinn[14], Anne Boland[14], Jean-François Deleuze[14,15], Lauriane Kuhn[16], Johana Chicher[16], Philippe Hammann[16], Sylvie Friant[6] (iD), Christelle Etard[2], Elke Krüger[3,*] (iD), Jean Muller[1,5,**] (iD), Uwe Strähle[2,***] & Hélène Dollfus[1,4,9,****] (iD)

## Abstract

The ubiquitin–proteasome system degrades ubiquitin-modified proteins to maintain protein homeostasis and to control signalling. Whole-genome sequencing of patients with severe deafness and early-onset cataracts as part of a neurological, sensorial and cutaneous novel syndrome identified a unique deep intronic homozygous variant in the *PSMC3* gene, encoding the proteasome ATPase subunit Rpt5, which lead to the transcription of a cryptic exon. The proteasome content and activity in patient's fibroblasts was however unaffected. Nevertheless, patient's cells exhibited impaired protein homeostasis characterized by accumulation of ubiquitinated proteins suggesting severe proteotoxic stress. Indeed, the TCF11/Nrf1 transcriptional pathway allowing proteasome recovery after proteasome inhibition is permanently activated in the patient's fibroblasts. Upon chemical proteasome inhibition, this pathway was however impaired in patient's cells, which were unable to compensate for proteotoxic stress although a higher proteasome content and activity. Zebrafish modelling for knockout in *PSMC3* remarkably reproduced the human phenotype with inner ear development anomalies as well as cataracts, suggesting that Rpt5 plays a major role in inner ear, lens and central nervous system development.

**Keywords** cataract; deafness; neurosensory disease; proteasome; *PSMC3*
**Subject Categories** Genetics, Gene Therapy & Genetic Disease; Neuroscience

## Introduction

Early-onset deafness is one of the most common causes of developmental disorder in children (prevalence rate of 2–4/1,000 infants),

1  Laboratoire de Génétique Médicale, INSERM, UMRS_1112, Institut de Génétique Médicale d'Alsace (IGMA), Université de Strasbourg, Faculté de médecine de Strasbourg, Strasbourg, France
2  Karlsruhe Institute of Technology (KIT), Institut für Chemische und Biologische Systeme (IBCS, BIP), Eggenstein-Leopoldshafen, Germany
3  Institut für Medizinische Biochemie und Molekularbiologie (IMBM), Universitätsmedizin Greifswald, Greifswald, Germany
4  Service de Génétique Médicale, Hôpitaux Universitaires de Strasbourg, Strasbourg, France
5  Laboratoires de Diagnostic Génétique, Hôpitaux Universitaires de Strasbourg, Strasbourg, France
6  Laboratoire de Génétique Moléculaire, Génomique, Microbiologie (GMGM), UMR7156, Centre National de Recherche Scientifique (CNRS), Université de Strasbourg, Strasbourg, France
7  Laboratory of Molecular and Developmental Biology, National Institute of Genetics, Mishima, Japan
8  Department of Genetics, SOKENDAI (The Graduate University for Advanced Studies), Mishima, Japan
9  Filière SENSGENE, Centre de Référence pour les affections rares en génétique ophtalmologique, CARGO, Hôpitaux Universitaires de Strasbourg, Strasbourg, France
10  Service de chirurgie ORL, Hôpitaux Universitaires de Strasbourg, Strasbourg, France
11  Department of Ophthalmology, Hôpitaux Universitaires de Strasbourg, Strasbourg, France
12  Service de Pédiatrie, Hôpitaux Universitaires de Strasbourg, Strasbourg, France
13  Faculté de Médecine, Hôpitaux Universitaires, Université de Strasbourg et Clinique Dermatologique, Strasbourg, France
14  Centre National de Recherche en Génomique Humaine (CNRGH), Institut de Biologie François Jacob, CEA, Université Paris-Saclay, Evry, France
15  Centre d'études du polymorphisme humain-Fondation Jean Dausset, Paris, France
16  CNRS FRC1589, Institut de Biologie Moléculaire et Cellulaire (IBMC), Plateforme Protéomique Strasbourg-Esplanade, Strasbourg, France
  *Corresponding author. Tel: +49 3834 865400; E-mail: elke.krueger@uni-greifswald.de
  **Corresponding author. Tel: +33 369550777; E-mail: jeanmuller@unistra.fr
  ***Corresponding author. Tel: +49 72160828327; E-mail: uwe.straehle@kit.edu
  ****Corresponding author. Tel: +33 368853341; E-mail: dollfus@unistra.fr
  †These authors contributed equally to this work

and identically, early-onset cataract is the most important cause of paediatric visual impairment worldwide (prevalence form 2–13.6/10,000 according to regions) accounting for 10% of the causes of childhood blindness. Each condition can be attributed to environmental causes (intrauterine infections, inflammation, trauma or metabolic diseases) or to genetic causes with a well-recognized very high level of genetic heterogeneity with 59 known genes causing early-onset cataracts and 196 genes known to cause severe deafness (Azaiez *et al*, 2018; Reis & Semina, 2018). Patients presenting both entities simultaneously, early-onset severe deafness and congenital cataracts, are thought to be mainly due to teratogenic exposure during pregnancy especially infections and are, nowadays, considered to be very rare. Indeed, only very few genetic inherited entities associating both congenital cataracts and deafness have been reported so far. The Aymé-Gripp syndrome (cataract, deafness, intellectual disability, seizures and Down syndrome like facies) has been recently linked to *de novo* pathogenic variants in the *MAF* gene, a leucine zipper-containing transcription factor of the AP1 superfamily (Niceta *et al*, 2015). In addition, dominant pathogenic variants in *WFS1* (recessive loss-of-function variants are responsible for Wolfram syndrome) have been described in children with congenital cataracts and congenital deafness presenting in the context of neonatal/infancy-onset diabetes (De Franco *et al*, 2017).

Herein, using whole-genome sequencing, we describe a novel homozygous non-coding pathogenic variant in *PSMC3* associated with severe congenital deafness and early-onset cataracts and various neurological features in three patients from a very large consanguineous family. *PSMC3* encodes the 26S regulatory subunit 6A also known as the 26S proteasome AAA-ATPase subunit (Rpt5) of the 19S proteasome complex responsible for recognition, unfolding and translocation of substrates into the 20S proteolytic cavity of the proteasome (Tanaka, 2009). The proteasome is a multiprotein complex involved in the ATP-dependent degradation of ubiquitinated proteins to maintain cellular protein homeostasis and to control the abundance of many regulatory molecules. The 26S proteasome consists of two complexes: a catalytic 28-subunit barrel shaped core particle (20S) that is capped at the top or the bottom by one 19 subunit regulatory particle (19S). The core particle contains the catalytic subunits β1, β2 and β5 exhibiting caspase-, trypsin- and chymotrypsin-like activities, respectively. Recognition of a substrate with the requisite number and configuration of ubiquitin is mediated principally by both Rpn10 and Rpn13 subunits, which act as ubiquitin receptors (Deveraux *et al*, 1994; Husnjak *et al*, 2008). To allow substrate degradation, ubiquitin is first removed by Rpn11, a metalloprotease subunit in the lid (Yao & Cohen, 2002). The globular domains of a substrate are then unfolded mechanically by a ring-like heterohexameric complex consisting of six distinct subunits, Rpt1 to Rpt6, which belong to the ATPases associated with diverse cellular activities (AAA) family (Chen *et al*, 2016). PSMC3 encodes for Rpt5 involved in the substrate unfolding and translocation, which are then presumably catalysed (Lam *et al*, 2002; Tanaka, 2009).

In mammalian cells, a major compensation mechanism for proteasome dysfunction is governed by the ER membrane-resident TCF11/Nrf1 protein (Radhakrishnan *et al*, 2010; Steffen *et al*, 2010; Sotzny *et al*, 2016). Typical stimuli for TCF11/Nrf1 activation include proteasome inhibition and/or impairment, which results in the release of C-terminal processed TCF11/Nrf1 fragment from the ER membrane following a complex series of molecular events involving the enzymes NGLY1 and DDI2. The cleaved TCF11/Nrf1 fragment enters then into the nucleus and acts as a transcription factor to promote the expression of ARE-responsive genes including 19S and 20S proteasome subunits, thereby augmenting the pool of proteasomes so that protein homeostasis can be preserved (Radhakrishnan *et al*, 2010; Steffen *et al*, 2010; Sotzny, *et al*, 2016).

We suggest that biallelic loss of PSMC3 causes a novel autosomal recessive syndrome with varying degrees of neurosensorial dysfunctions including the combination of cataract and deafness. Functional analysis of patient's cells revealed that although normal amount of proteasome proteins can be observed in steady-state conditions, the cells are unable to adapt to proteotoxic stress. The use of zebrafish morpholinos and CRIPSR-Cas9 assays confirmed the same combination of sensory phenotypes upon inactivating PSMC3 expression.

# Results

## Patient phenotypes

Three patients with a novel syndromic neurosensory-cutaneous presentation consulted independently to our clinical centre over a period of 15 years. Careful analysis revealed that they originated from the same small village (Amarat) in the Kayseri region of Turkey and belong to the same large extended consanguineous family (Fig 1A). The proband is a male individual (II.4) diagnosed at the age of 8 months with profound perceptive deafness and subsequently benefited from a cochlear implantation. He was referred at the age of 2 years old to our centre because of visual impairment due to bilateral cataracts for which he underwent bilateral lensectomies. With years, he developed severe developmental delay and severe intellectual deficiency (no words, limited comprehension). Several facial features were noted (Table 1 and Fig 1B). In addition, severe autistic features were revealed at the age of 2.5 years old (Table 1). A full metabolic exploration was normal. At the age of 5, he developed subcutaneous deposits at the level of the knees and elbows (Fig 1C). At the age of 10, he developed white hair at the level of the two legs as opposed to the dark pigmented hair on the rest of the body. More recent examination revealed also a peripheral polyneuropathy of lower limbs. The two other patients (II.2 and II.7) were referred at the age of 1 year old and share the same severe perceptive deafness (for which they also benefited from a cochlear implantation), visual impairment due to bilateral obstruent cataracts (for which they also had bilateral lensectomies) and subcutaneous deposits. Patient II.2 did not present with autistic features but had moderate developmental delay (able to read and write few words, but no understanding of complex sentences) and a significative polyneuropathy of the lower limbs (more pronounced that II.4). Patient II.7 did not present with polyneuropathy as opposed to the other patient, but like patient II.4, he presented severe developmental delay with autistic features. For each patient, otoacoustic emissions were positive at birth. However, deafness was suspected for all of them within the early months of life, respectively, 8 months (II.4) and 1 year and 3 months (II.2, II.7). Auditory brainstem response was in favour of profound deafness (no response at 110 dB). The MRI did not reveal any anomalies of the

cochleovestibular nerves or labyrinthitis. However, the temporal bone CT scan analysis of patient II.7 revealed lateral semicircular canal malformations with the absence of the bony island in the right ear and a small bony island in the left ear (Fig 1D).

## Identification of a rare deep intronic variation in *PSMC3*

During the years of follow-up, each patient was explored for known deafness and cataract genes by Sanger sequencing (in particular for

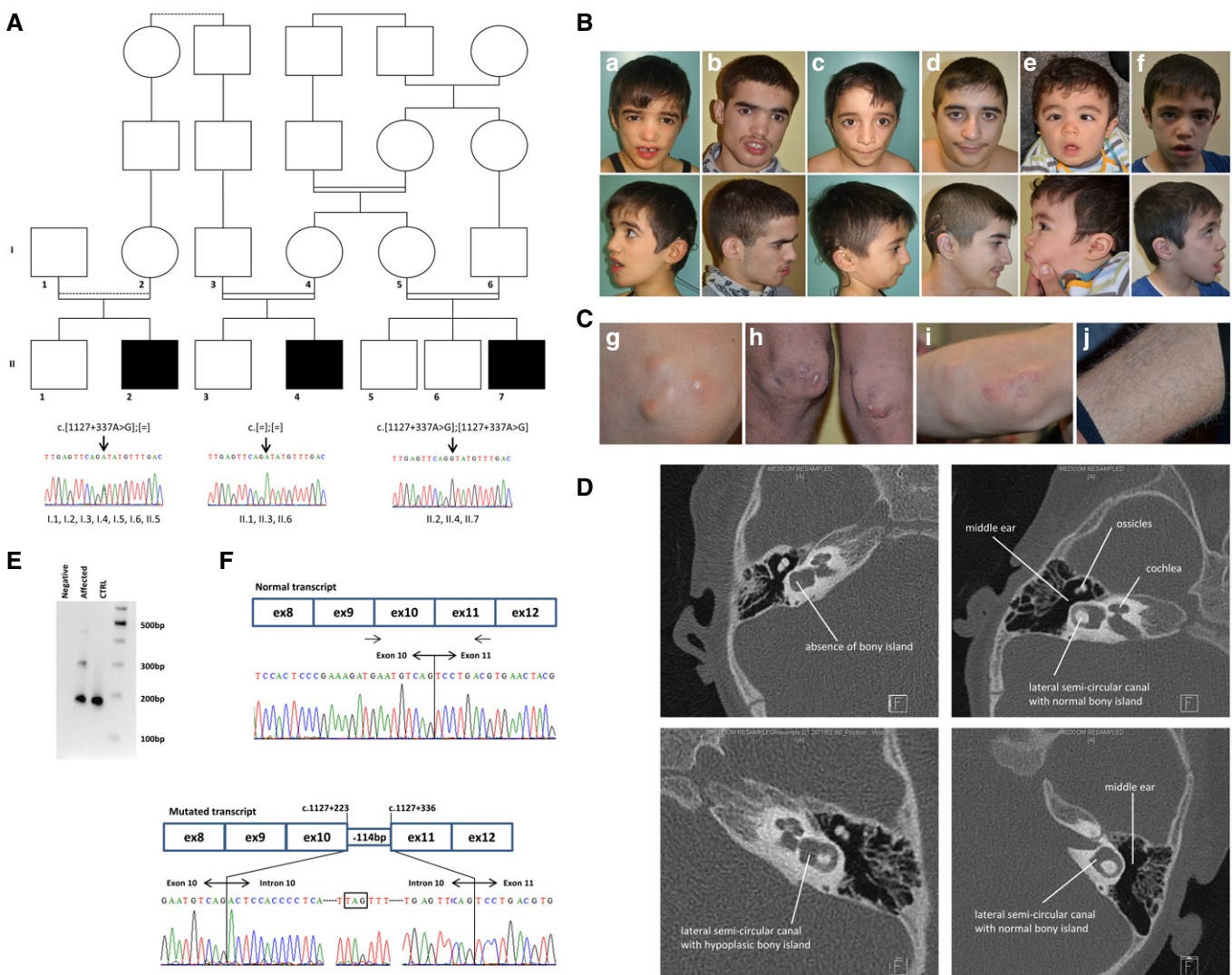

**Figure 1. Family pedigree and cDNA analysis.**

A   Family pedigree. Variant segregation analysis of *PSMC3*. Electropherogram of a part of intron 10 of *PSMC3* encompassing the identified variation (c.[1127 + 337A>G]; [1127 + 337A > G], p.[(Ser376Arg15*)];[(Ser376Arg15*)]) in the affected individuals, their unaffected parents and siblings. The variation was found at the homozygous state in the affected individuals (II.2, II.4, II.7) and at the heterozygous state in the parents (I.1, I.2, I.3, I.4, I.5, I.6) and was either at the heterozygous state (II.5) or absent in the unaffected siblings (II.1, II.3, II.6).

B   Face (up) and profile (down) photographs for patients II.4 (a: 8 yo, b: 16 yo), II.2 (c: 6 yo, d: 14 yo) and II.7 (e: 1 yo, f: 7 yo) over time. One can observe prominent supraciliary arches, synophrys, sunken cheeks, short philtrum and retrusion in the malar region.

C   Subcutaneous calcifications found only on knees (g: 9 yo and h: 16 yo) and on elbows (i: 9 yo) of patient II.4. White hair were present only on the legs of the 3 patients as illustrated for patient II.4 (j: 16 yo).

D   Temporal bone CT scan from patient II.7 (left column) and a control (right column) showing malformation of the semicircular canal. The left ear is shown on the upper panels while the right ear on the lower panels.

E   Amplification of the cDNA fragment between exons 9–10 and 11 of *PSMC3* showing the abnormally spliced RNA fragment. One band at 180 bp representing the normal allele is seen for the control and two bands for the individual II.4 (pathologic allele at 300 bp).

F   Schematic representation for the incorporation of the 114 bp intronic sequence resulting from the c.1127 + 337A > G deep intronic variation on the mRNA. Sanger sequencing of the fragment between exons 9–10 and 11 of the *PSMC3* cDNA obtained from patient II.4 fibroblasts' RNA, showing the insertion of the 114 bp cryptic exon. As a comparison, the schematic representation and sequence from a control individual are shown above.

Source data are available online for this figure.

*GJB2*, one of the patient being an heterozygous carrier of the c.30delG well-known recurrent pathogenic variant) but also using larger assays such as whole-exome sequencing (WES) with a specific focus on known deafness and cataract genes (Appen dix Table S1) and standard chromosomal explorations (karyotype and chromosomal microarray analysis), but all were negative (see Appendix Supplementary Methods). Considering that affected individuals may harbour pathogenic variants in a region not covered by the WES (i.e. intronic and intergenic) or not well detected (i.e. structural variations) (Geoffroy *et al*, 2018b), we applied whole-genome sequencing (WGS) to the three affected individuals (II.2, II.4 and II.7) and two healthy individuals (II.1 and II.3). Given the known consanguinity in the family, our analysis was focused on homozygous variations and more specifically within homozygous regions defined by the SNP arrays (Appendix Fig S1). In addition to the classical filtering strategy including functional criteria, frequency in population-based databases and cosegregation analysis (see Materials and Methods), we defined a list of 4,846 potentially interacting genes with the already known human cataract (59) and deafness genes (196) (Appendix Fig S2). This strategy allowed us to identify from the ~5,000,000 variations per WGS, six variations out of which a unique homozygous variant in the intron 10 of the *PSMC3* gene (c.1127 + 337A>G, p. (?)) remained of interest (Appendix Table S2 and Fig S3). This variant was not present in any variation database (e.g. gnomAD) and is predicted to create a new donor splice site. Interestingly, among others PSMC3 was shown to interact (Appendix Tables S3 and S4, and Fig S4) with CHMP4B (MIM 610897), ACTG1 (MIM 102560) and GJB6 (MIM 604418) involved in cataract and deafness.

## Effect of the variation on *PSMC3* expression and localization

In order to assess the effect of this deep intronic variation, we investigated the expression of the gene in the patient's fibroblasts. The suspected new donor site could be associated with multiple acceptor sites within intron 10 (Appendix Fig S5). RNA analysis revealed an additional band specific to the affected patient that was further explored by Sanger sequencing (Fig 1E). The consequence of this variation is the inclusion of a cryptic 114 bp exon during the splicing process based on the intronic sequence (r.1127_1127+1insACTC-CACCCCTCATCTGAAGGCACAGAGGCTGGAGGCACTTAGTTTCCT GGCCTCACACCTCAGCCCATTAACACACGCCAGGAATGGCCGGGAC CAGATGGACTTGAGTTCAG) (Fig 1F) that is predicted to add 15aa (LeuHisProSerSerGluGlyThrGluAlaGlyGlyThr) at position 376 followed by a stop (p.(Ser376Argfs15*)). Analysis at the RNA level showed a significantly reduced level of *PSMC3* mRNA as well as the presence of an additional truncated form (Appendix Fig S6). However, no difference in expression or localization of the PSMC3 protein could be detected between the control and the patient cells under normal condition, indicating that the truncated form is probably not stable (Appendix Fig S7).

## Functional effect of the PSMC3 variant to the proteasome function and assembly

Given the role of PSMC3 in protein degradation, we determined the intracellular level of ubiquitinated proteins in patient cells compared to controls. Our results show an increased level of ubiquitinated proteins in patient cells (Fig 2A and B), suggesting that the proteasomal proteolysis is less efficient. Having shown a possible effect on proteasome function, we next investigated how the variant could affect the proteasome assembly and dynamics. First, in standard condition, PSMC3 protein and its partners were immunoprecipitated from either controls' or patient's fibroblasts and revealed by mass spectrometry (Fig 2C). The PSMC3 protein was detected in the input control and patient, indicating that the variant does not affect the protein stability of the remaining wild-type allele confirming the Western blot analysis. Looking at the interacting partners, one can notice that each proteasome subunit could be detected revealing no apparent defect in the general organization of the proteasome. However, protein abundance of each proteasome subcomplex estimated from the number of mass spectrometry spectra observed between the controls and the patient (Appendix Table S5) shows a general increase of the proteasome subunits (approximately 20%). Looking more specifically at each subcomplex, the increase is mainly due to the core particle including the alpha and beta subunits with 1.5-fold for all PSMA proteins and most PSMB protein and even a 2.0-fold increase for PSMB2/4/6, while PSMC and PSMD remain at the same ratio.

To further characterize the consequences of the *PSMC3* variant on the functionality of the ubiquitin–proteasome system (UPS), cell lysates derived from control and patient primary fibroblasts were analysed by non-denaturing PAGE with proteasome bands being visualized by their ability to hydrolyse the Suc-LLVY fluorogenic peptide. As shown in Fig 3A, gel overlay assay for peptidase activity revealed two strongly stained bands corresponding to the positions of the 20S and 26S proteasome complexes, respectively. However, no discernible differences could be detected in the chymotrypsin-like activity of both 20S and 26S complexes between control and patient cells, suggesting that peptide hydrolysis in the 20S proteolytic core is not substantially impaired by the *PSMC3* pathogenic variant. The notion that proteasome activity does not vary between these two samples was further confirmed by monitoring the degradation rate of the Suc-LLVY peptide directly in whole-cell extracts from control and patient fibroblasts over a 180-min period that was almost identical in both samples (Fig 3B). In order to characterize the proteasome populations in cells carrying the deep intronic *PSMC3* homozygous variant, samples separated by non-denaturing PAGE were subsequently analysed by Western blotting. As expected, using an antibody against the proteasome subunit α6, two major bands were observed in the 20S and 26S regions (Fig 3C). Interestingly, the signal intensity for both proteasome complexes was significantly stronger in patient fibroblasts, suggesting that the amount of intracellular proteasome pools in these cells was higher than those of control fibroblasts. Western blotting against the PSMC3 subunit revealed two bands in the 26S proteasome area and corresponding to single and double-capped proteasomes (19S-20S and 19S-20S-19S, respectively) and confirmed the higher amount of these complexes in patient cells (Fig 3C); however, there are some lower bands corresponding to 19S precursor intermediates, indicating that assembly of 19S complexes is affected. As shown in Fig 3C, staining for PA28-α, a subunit of the alternative proteasome regulator PA28, revealed one major band corresponding to the position of the 20S proteasome, indicating the 20S proteasomes in these samples mainly consist of PA28-20S complexes. Again, patient fibroblasts exhibited a stronger expression level of such homo-PA28

**Table 1. Clinical description of the patients with *PSMC3* pathogenic variants.**

| | II.4 | II.2 | II.7 |
|---|---|---|---|
| Birth date | 24/12/2003 | 02/02/2005 | 15/03/2012 |
| **Neurosensorial features** | | | |
| Congenital cataract | + | + | + |
| Strabismus | + | − | + |
| Congenital deafness | + | + | + |
| **Facial dysmorphism features** | | | |
| Round ears | + | − | − |
| Synophrys | + | − | − |
| Short philtrum | + | + | − |
| Malar region retrusion | + | + | + |
| Prominent supraciliary arches | + | + | + |
| Sunken cheeks | + | + | + |
| Preauricular fibrochondroma | − | − | + |
| Thin upper lip | + | + | − |
| **Neurologic features** | | | |
| Developmental delay | S | M | S |
| Autistic features | + | − | + |
| Peripheral polyneuropathy of lower limbs | + | + | − |
| **Cutaneous features** | | | |
| Calcifications of elbows and knees | + | + | + |
| Depigmented hairs of lower limbs | + | + | − |

M, moderate; S, severe.

complexes than their wild-type counterparts. To validate the notion that the homozygous pathogenic variant results in increased assembly of newly synthetized proteasome complexes, patient fibroblasts were compared to control ones for their content in various proteasome subunits using SDS–PAGE followed by Western blotting. As illustrated in Fig 3D, the steady-state expression level of most of the β, α and Rpt subunits was substantially higher in the patient's cells. A minor band corresponding to the expected size of the truncated RPT5 variant (i.e. 43,548.95 Da) was observed in the patient's sample (Fig 3D, long exposure time). Unexpectedly, the increased proteasome content was accompanied by a parallel rise of ubiquitin-modified proteins in these cells, suggesting that proteasomes from patients bearing the homozygous *PSMC3* pathogenic variant, although being in greater number, are ineffective. Altogether, these results point to a defective proteasome function in subjects carrying the deep intronic homozygous *PSMC3* pathogenic variant, which seems to be compensated by an ongoing assembly of newly synthetized 20S and 26S complexes.

We next sought to determine the impact of this variant on the ability of the cells to respond to perturbed protein homeostasis following proteasome dysfunction. To this end, both control and patient cells were subjected to a 16-h treatment with the β5/β5i-specific inhibitor carfilzomib in a non-toxic concentration prior to SDS–PAGE and Western blotting analysis using various antibodies specific for proteasome subunits. As shown in Fig 4, *PSMC3* mutant cells were endowed with higher amounts of immunoproteasome subunits and proteasome activator PA28-α in untreated conditions when compared to their wild-type counterparts. Most importantly, control cells exposed to carfilzomib could successfully compensate the applied proteotoxic stress by increasing their pools of intracellular proteasomes, as evidenced by elevated expression of all investigated β- and Rpt subunits. As expected, this process was preceded by the processing of the ER membrane-resident protein TCF11/Nrf1 (Fig 4), which is the transcription factor acting on nuclear genes encoding 19S and 20S proteasome subunits (Radhakrishnan *et al*, 2010; Steffen *et al*, 2010; Sotzny *et al*, 2016). Strikingly, the level of processed TCF11/Nrf1 in response to carfilzomib was much lower in cells carrying the homozygous *PSMC3* pathogenic variant than that observed in control cells. Accordingly, the patient's fibroblasts were unable to upregulate their proteasome subunits following proteasome inhibition, as determined by decreased expression levels of proteasome subunits and PA28-α (Fig 4).

To confirm that the cellular phenotype of patient fibroblasts was due to the homozygous *PSMC3* variant, we next conducted rescue experiments in which these cells were subjected to 24-h transfection with an expression vector encoding wild-type *PSMC3*. As shown in Fig 5A, patient cells overexpressing intact *PSMC3* exhibited a diminished accumulation of K48-linked ubiquitin–protein conjugates when compared to control cells or empty-vector cells, as determined by Western blotting. Densitometry analysis of the ubiquitin band intensities reveals that the pool of ubiquitin-modified proteins was significantly reduced by about 20% in *PSMC3*-rescued patient fibroblasts at 24-h post-transfection (Fig 5B). In addition, introducing wild-type *PSMC3* into patient cells resulted in a decreased activation of TCF11/Nrf1, as evidenced by the absence of the TCF11/Nrf1 processed form under normal conditions (Fig 5C). Most importantly, overexpression of wild-type *PSMC3* successfully prevented the loss of proteasome subunits and PA28-α in patient fibroblasts following carfilzomib treatment (Fig 5C). Altogether, these data clearly identified the homozygous *PSMC3* mutant as the genetic cause of proteasome failure affecting protein homeostasis under stress conditions.

## Effect of PSMC3 loss of function in zebrafish similar to patients' phenotype

To establish a functional link between the observed decreased proteasome activity and the phenotype observed in the patients, we next assessed the lens and the ear in the zebrafish model. The zebrafish orthologue (Ensembl, ENSDARG00000007141) of human *PSMC3* is located on chromosome 7 with two predicted protein-coding splice variants (404 and 427 amino acids). Both zebrafish *psmc3* isoforms share 83% sequence identity with the human orthologue. We confirmed by *in situ* hybridization that *psmc3* is maternally expressed and not spatially restricted (Appendix Fig S8; Data ref: Thisse & Thisse, 2004). Injection of morpholinos against *psmc3* generated zebrafish morphants embryos that were examined at 4 days post-fertilization (dpf) for lens or ear abnormalities. The lens size of morphants was slightly smaller than in control or uninjected embryos (Appendix Fig S9A). For cataract detection, we used a protocol based on confocal reflection microscopy, a labelling-free non-invasive imaging method that enables the detection of abnormal light reflection in the lens of living embryos (Fig 6A; Takamiya

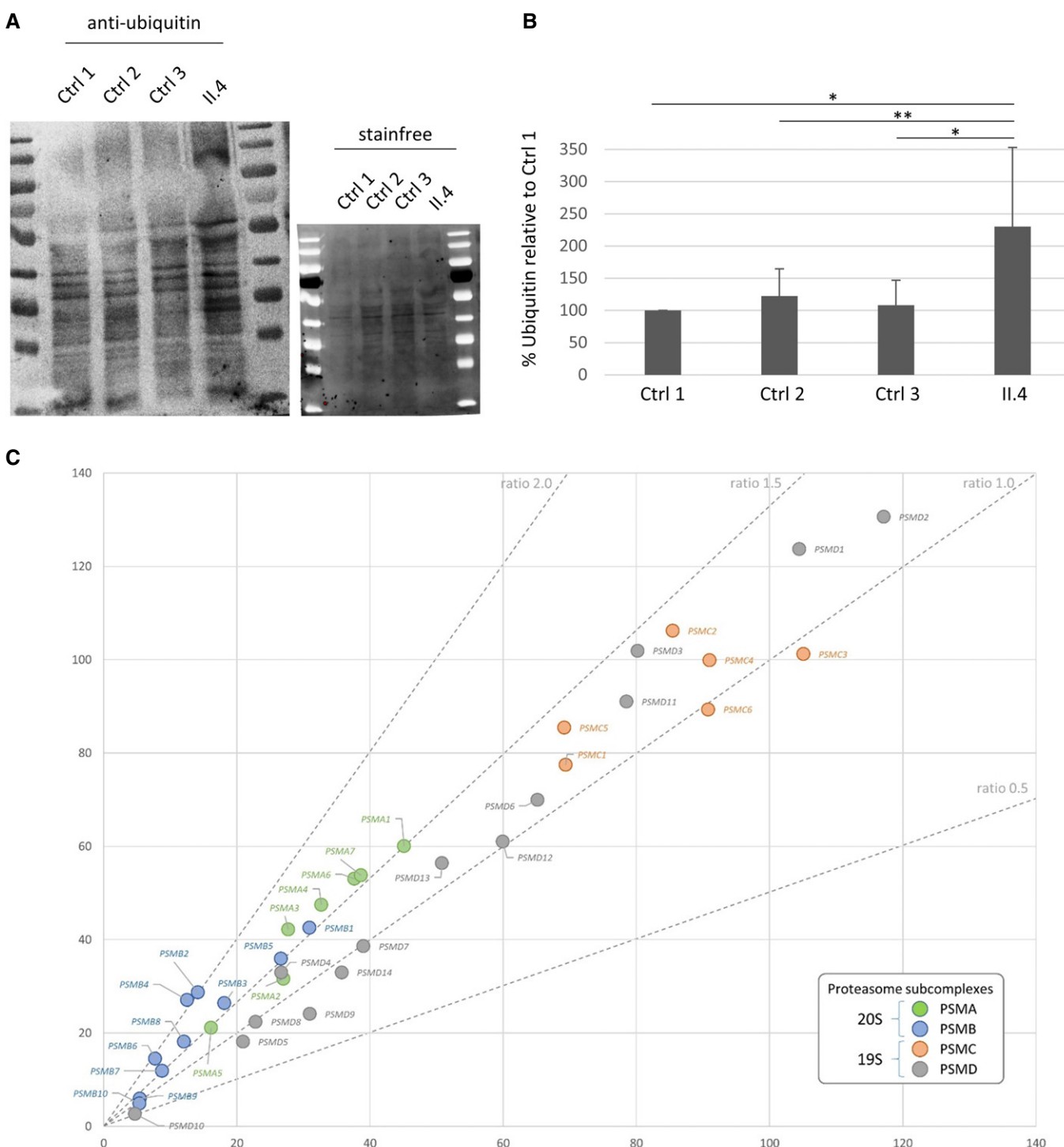

**Figure 2. Effect of the deep intronic *PSMC3* variant on the proteasome function.**

A  Anti-ubiquitin Western blot in control and patient fibroblasts and total amount of proteins loaded (stainfree) showing increased ubiquitination in patient cells (lane 4).

B  Histogram showing the quantification of ubiquitin with Western blot assays. The data shown correspond to the sum of all bands detected by the anti-ubiquitin antibody expressed as a percentage of the amount of ubiquitin in "Control 1" cells. Bars show mean of ten independent experiments ± SD ($n = 10$, $t$-test $*P < 0.01$, $**P < 0.05$).

C  Mass spectrometry results from the co-immunoprecipitation with PSMC3 are displayed as the normalized total number of spectra count of each protein computed as the mean from 3 controls ($x$ axis) vs. the mean of patient II.4 triplicate. Proteasome subcomplexes are coloured according to the displayed legend, and standard ratio lines are drawn.

et al, 2016). Morpholino injections revealed significant abnormal lens reflections in 95% of the morphants (n = 55), whereas only 2% of 5 bp mismatched morpholino-injected controls (n = 45) and none of the uninjected embryos (n = 20) showed cataract (Fig 6B and B′). The observed cataract was not due to increased apoptosis, as TUNEL staining did not reveal more positive nuclei in the morphant compared to wild-type embryos (Appendix Fig S10).

As deafness has been reported for all three human patients, we subsequently examined the inner ear development of 4 dpf zebrafish morphants (Fig 6C). psmc3 morphants displayed a smaller ear

compared to control or uninjected embryos (Appendix Fig S9C). Interestingly, the majority of morphants presented anomalies during the semicircular canal morphogenesis. While the epithelium projections of all uninjected (n = 20) and all control-injected (n = 45) embryos were fused and formed pillars after 4 dpf, the canal projections of morphants failed to fuse in 79% of the cases (n = 58, Fig 6D and D′). The specificity of the morpholino was confirmed by a rescue experiment by co-injecting the full-length splice morpholino-resistant zebrafish psmc3 mRNA. The cataract phenotype was rescued in ∼ 58% of the cases (n = 60), while the ear phenotype

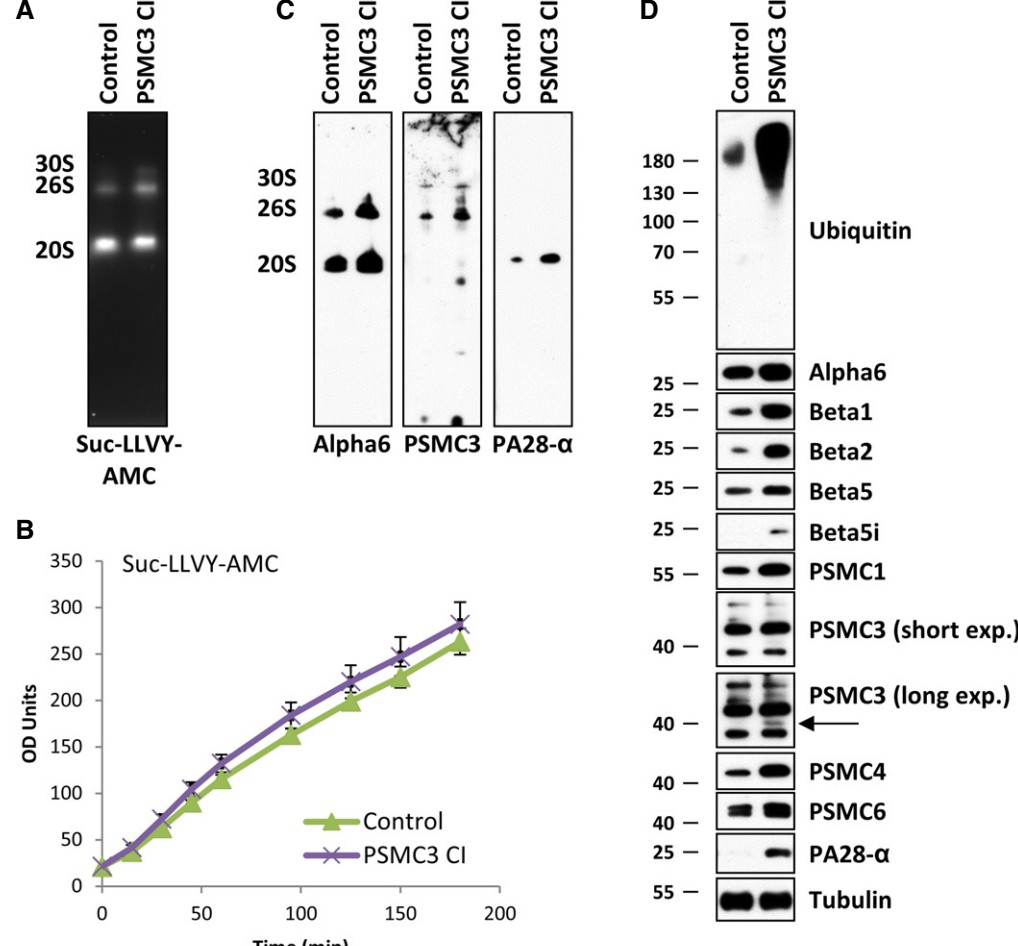

**Figure 3. Fibroblasts derived from patient carrying the c.1127 + 337A>G homozygous *PSMC3* variation exhibit an increased amount of both proteasome complexes and ubiquitin–protein conjugates.**

A   Whole-cell lysates from control and patient (case index, CI) fibroblasts were assessed by 3–12% native-PAGE gradient gels with proteasome bands (30S, 26S and 20S complexes) visualized by their ability to cleave the Suc-LLVY-AMC fluorogenic peptide.

B   Ten micrograms of control and patient cell lysates was tested for their chymotrypsin-like activity by incubating them with 0.1 mM of the Suc-LLVY-AMC substrate at 37°C over a 180-h period of time in quadruplicates on a 96-well plate. Indicated on the y-axis are the raw fluorescence values measured by a microplate reader and reflecting the AMC cleavage from the peptide. Bars show the mean of 4 independent experiments ± SD.

C   Proteasome complexes from control and patient fibroblasts separated by native-PAGE were subjected to Western blotting using antibodies specific for α6, Rpt5 (PSMC3) and PA28-α, as indicated.

D   Proteins extracted from control and CI PSMC3 were separated by 10 or 12.5% SDS–PAGE prior to Western blotting using primary antibodies directed against ubiquitin and several proteasome subunits and/or components including α6, β1, β; β5, β5i, Rpt2 (PSMC1), Rpt5 (PSMC3), Rpt3 (PSMC4), Rpt4 (PSMC6) and PA28-α, as indicated. For the PSMC3 staining, two exposure times are shown. Arrow indicates an additional PSMC3 species corresponding to the expected size of the truncated PSMC3 variant. Equal protein loading between samples was ensured by probing the membrane with an anti-α-Tubulin antibody.

Source data are available online for this figure.

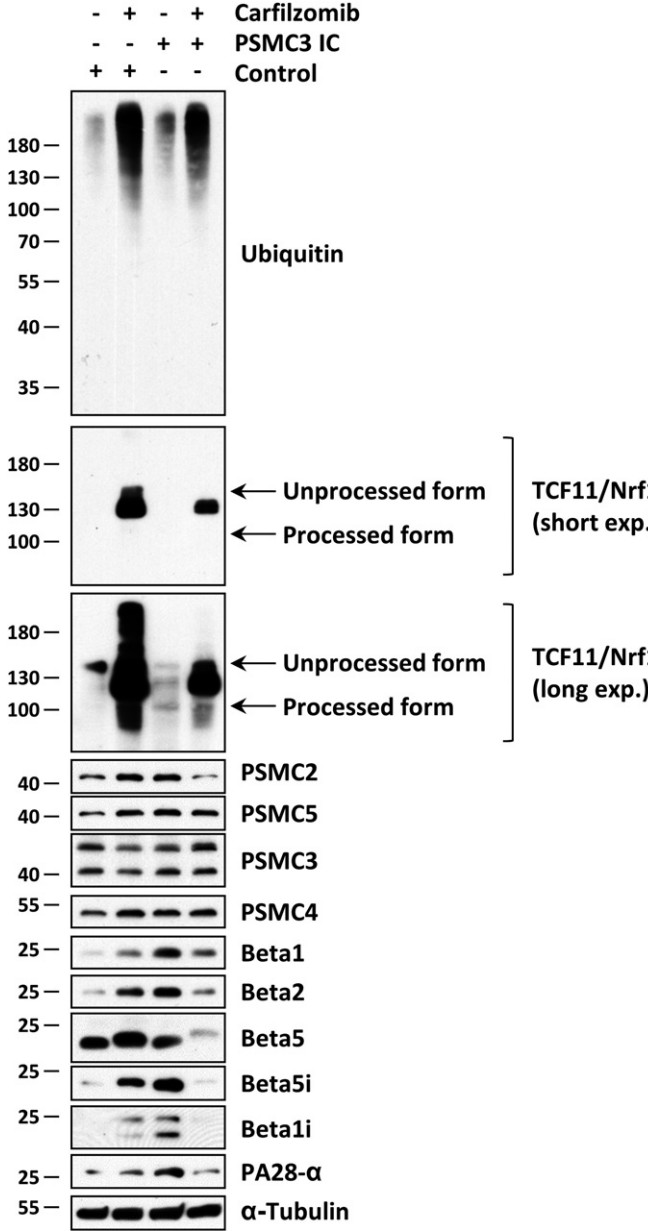

**Figure 4.  Patient fibroblasts carrying the c.1127 + 337A>G homozygous PSMC3 variation exhibit an exhausted TCF11/Nrf1 processing pathway, which prevents them to upregulate proteasome subunits in response to proteotoxic stress.**

Control and patient (index case, IC PSMC3) fibroblasts were exposed to a 16-h treatment with 30 nM of the proteasome inhibitor carfilzomib or left untreated (as a negative control). Following treatment, cells were collected and subjected to RIPA-mediated protein extraction prior to SDS–PAGE and subsequent Western blotting using antibodies specific for ubiquitin, TCF11/Nrf1, Rpt1 (PSMC2), Rpt3 (PSMC4), Rpt5 (PSMC3), Rpt6 (PSMC5), β1, β2, β5, β5i, β1i, PA28-α and α-Tubulin (loading control) as indicated. For the TCF11/Nrf1 staining, two exposure times are shown.

Source data are available online for this figure.

was rescued in 76% of the cases ($n = 60$; Fig 6B′ and D′). The outgrowth of the epithelial projections of the developing anterior and posterior semicircular canals begins around 48 hpf. From 57 to 68 hpf, the projections fuse in the centre of the ear and form the pillars. This is followed by the outgrowth of the projection of the lateral semicircular canal around 57 hpf and completed by the fusion with the other two pillars in the centre around 70 hpf (Geng et al, 2013). To analyse the semicircular canal morphogenesis in psmc3 morphants, life imaging was performed between 56 and 72 hpf in the transgenic line gSAIzGFF539A expressing a GFP signal in all three pillars. In all morphants ($n = 9$), the projections failed to fuse and form pillars during the observed time frame. In two morphants (22%), the outgrowth of epithelial projections even failed completely. In contrast, the projections of uninjected ($n = 2$) and control-injected ($n = 2$) embryos fused during the observed period in the centre of the ear and formed canal pillars (Movie EV1). Previous studies showed that a reduced number or a smaller size of otoliths, crystal-like structures required for the transmission of mechanical stimuli to the hair cells, can lead to deafness and balancing difficulties in zebrafish (Han et al, 2011; Stooke-Vaughan et al, 2015). psmc3 morphants did not present any otolith defect (Appendix Fig S11A–C). In addition, expression of otopetrin, a gene required for proper otolith formation, was unaffected at 28 hpf and 4 dpf (Appendix Fig S9D).

As autism has been reported for one of the patients and brain malformation has been reported previously in some autistic patients and autistic zebrafish morphants (Elsen et al, 2009), we investigated possible morphological brain changes in psmc3 morphants. In situ hybridization targeting brain markers such as krox20, msxc, her8a and sox19b was performed on 24 hpf embryos but did not reveal any obvious differences in their expression patterns (Appendix Fig S12).

To confirm these results, we additionally used the CRISPR/Cas9 system to knockdown psmc3 (Fig 6B–D). The high cutting efficiency of the CRISPR/Cas9 founders (F0) was evaluated to 55.2% (gRNA1) and 53.7% (gRNA2) (Appendix Fig S13) (Etard et al, 2017). CRISPR/Cas9 founders (i.e. crispants) are often genetically mosaics. However, in cases of highly efficient gRNAs, they have been shown to recapitulate mutant phenotypes successfully (Küry et al, 2017; Teboul et al, 2017; Paone et al, 2018). Both psmc3 crispants used in this study displayed a cataract phenotype (100% of gRNA1 + Cas9 ($n = 20$) and 95% of gRNA2 + Cas9 ($n = 20$), whereas none of the control embryos (injected with gRNA1 ($n = 20$) or gRNA2 ($n = 10$) without the Cas9 protein) showed abnormal lens reflections (Fig 6B and B′, and Appendix S13A and A′). Moreover, we recapitulated the ear phenotype seen in psmc3 morphants. While the ear pillars of uninjected ($n = 10$) and control-injected ($n = 20$) embryos formed after 4 dpf, the projections of both crispants failed to fuse in 65% of gRNA1 ($n = 10$)- and 93% of gRNA2 ($n = 30$)-injected embryos (Fig 6D and D′, and Appendix S13B and B′). Co-injection of gRNA2, Cas9 and the psmc3 mRNA also led to a partial rescue of the lens and ear phenotype with only 63% of abnormal lens reflection instead of 100% ($n = 19$) and 37% of unfused canal projections instead of 93% ($n = 30$; Fig 6D′). Performing an in situ hybridization examining the mRNA expression of versican a and versican b, two genes suggested to be required for a proper canal fusion event (Geng et al, 2013), significant differences could be observed after 72 hpf. Indeed, both genes were highly expressed in the whole ear tissue, whereas versican a was not expressed and versican b was restricted to the dorsolateral septum in wild-type or control-injected embryos (Appendix Fig S14B–D).

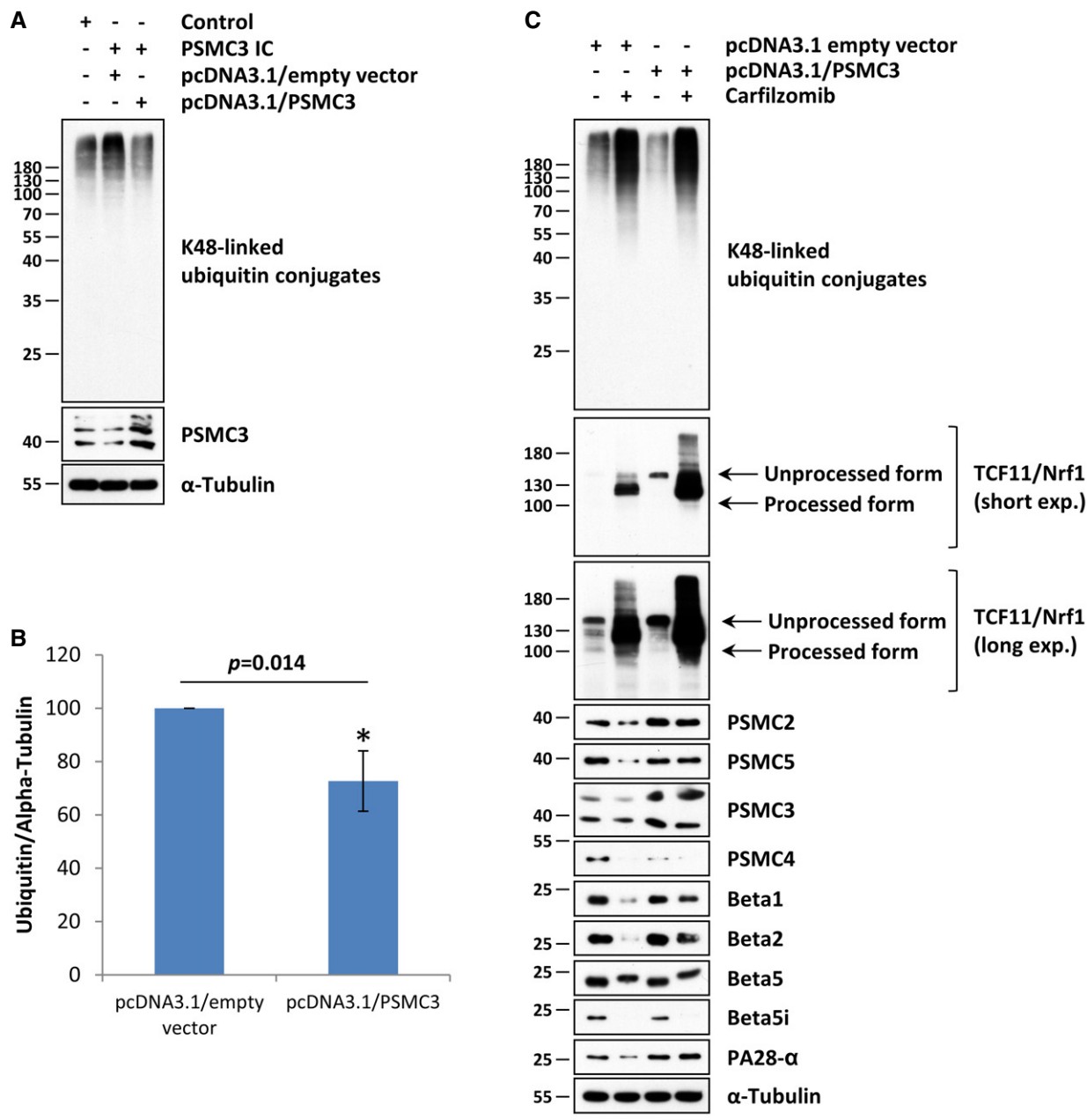

**Figure 5. Overexpression of wild-type *PSMC3* in patient fibroblast restores intracellular protein homeostasis and prevents the loss of proteasome subunit expression in response to proteasome inhibition.**

A Control and/or patient (index case, IC PSMC3) fibroblasts were subjected to a 24-h transfection with pcDNA3.1/empty vector (mock) or pcDNA3.1/PSMC3 prior to RIPA-mediated protein extraction and subsequent Western blotting using antibodies specific for ubiquitin, PSMC3 (i.e. Rpt5) and α-Tubulin (loading control).

B Densitometry analysis showing the relative ubiquitin contents detected by Western blotting in patient fibroblasts exposed to either pcDNA3.1/empty vector (mock) or pcDNA3.1/PSMC3, as indicated. The $y$-axis represents the per cent changes in densitometry measurements (of pixel intensities using ImageJ), which are set as 100% for cells transfected with the pcDNA3.1/empty vector (mock) at 24 post-transfection ($n = 4$, *$P < 0.05$, $t$-test). Bars show the mean $\pm$ SEM.

C Patient (index case, IC PSMC3) fibroblasts transfected with either pcDNA3.1/empty vector (mock) or pcDNA3.1/PSMC3 were exposed to a 30-nM treatment of carfilzomib or left untreated (as a negative control). After 16 h, cells were collected and subjected to RIPA-mediated protein extraction prior to SDS–PAGE and subsequent Western blotting using antibodies specific for ubiquitin, TCF11/Nrf1, Rpt1 (PSMC2), Rpt3 (PSMC4), Rpt5 (PSMC3), Rpt6 (PSMC5), β1, β2, β5, β5i, PA28-α and α-Tubulin (loading control), as indicated. For the TCF11/Nrf1 staining, two exposure times are shown.

Source data are available online for this figure.

The inner ear possesses hair cells to sense both vestibular and auditory stimuli. These apical structures consist of a bundle of villi-like structures called stereocilia and kinocilia, collectively referred to as a hair bundle. Because these cilia have been shown to play a key role at least in mechanosensation during development (Kindt *et al*, 2012), we immunostained crispants (sgRNA2

injected with Cas9) and control-injected embryos (sgRNA2 without Cas9) at 5 dpf using anti-acetylated tubulin antibody. In 40% of crispants ($n = 15$), a reduced number of cilia was observed while

their number in control-injected embryos ($n = 10$) was similar to that of uninjected embryos ($n = 10$) (Fig 6E+E'). In order to examine the morphology of hair cells themselves, we used FM1-43. This

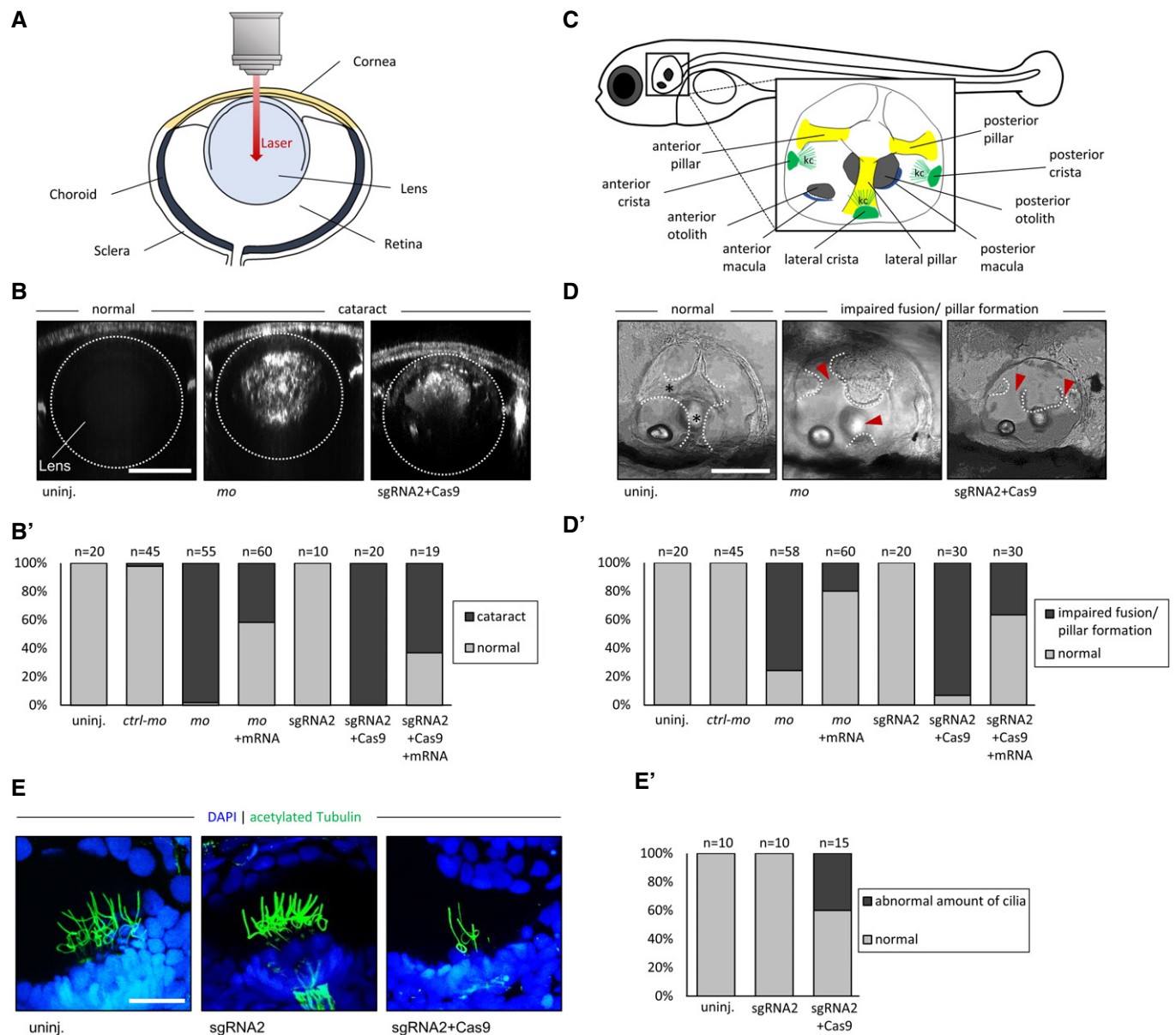

**Figure 6. *psmc3* morphants and F0 mosaic zebrafish exhibit cataract and show abnormalities during the semicircular canal development in the ear.**

A   Scheme of a zebrafish eye.

B, B'   Cataract detection revealed abnormal lens reflection in *psmc3* morpholino (MO)-mediated knockdown but not in controls (uninj, *ctrl-mo*). Similarly, abnormal lens reflection was also observed in embryos injected with sgRNA + Cas9 but not in sgRNA-injected embryos without Cas9 (sgRNA2). Co-injection of wt *psmc3* mRNA with either *psmc3-mo* or sgRNA2 + Cas9 reduced the number of embryos presenting abnormal lens reflection. Scale bar = 50 µm. (B') Quantification of embryos with abnormal lens reflection.

C   Representative image of a zebrafish ear at 4 dpf. kc = kinocilia.

D, D'   Brightfield images of inner ear development (lateral position). (D) Epithelial projections were fused and formed canal pillars in 4-day-old uninjected and control-injected fish (*ctrl-mo*, sgRNA2) but not in morphants (*mo*) and crispants (sgRNA2 + Cas9). Co-injection of wt *psmc3* mRNA with *psmc3-mo* or sgRNA + Cas9 reduced the number of embryos presenting abnormal ear phenotype. Black asterisks indicate fused pillars. Red arrowheads mark unfused projections. Scale bar = 100 µm. (D') Quantification of embryos with abnormal projection outgrowth.

E, E'   An anti-acetylated tubulin antibody (green) staining revealed an abnormal amount of kinocilia in *psmc3* crispants (sgRNA2 + Cas9) compared to uninjected and control-injected embryos (sgRNA2). Nuclei are stained in blue with DAPI. Representative images show kinocilia of the lateral cristae. Scale bar = 20 µm. (E') Quantification of embryos with an abnormal amount of kinocilia.

did not reveal any obvious difference between wild-type and morphant embryos. However, we observed that the length of the cilia was decreased when compared to wild-type embryos (Appendix Fig S15).

Taken together, these zebrafish assays confirm that *psmc3* plays a very important role in the development of a transparent lens and the semicircular canals of the inner ear—reminiscent of the human phenotype described herein.

## Discussion

In this study, we describe a novel homozygous deep intronic splice variant, identified in three patients with an unusual neurosensorial disease combining early-onset deafness, cataracts and subcutaneous deposits, in *PSMC3* encoding one of the proteasome subunit. Clinical data and functional analysis in patient's cells and zebrafish proved the effect of the variation and the consequences leading to a recessive form of proteasome deficit with a haploinsufficiency mechanism. The observation of a neurosensory disease broadens the spectrum of ubiquitin–proteasome system (UPS)-related disorders. Sequencing the entire genome of patients gives access to the whole spectrum of their variations and possibly disease-causing ones. WGS is a powerful tool (Belkadi *et al*, 2015) helping to identify variations not covered or missed by WES such as structural variations (Geoffroy *et al*, 2018b) or deep intronic variations (Vaz-Drago *et al*, 2017). Interestingly, in this study, we combined to WGS, homozygosity mapping and *in silico* predicted interactors to narrow down to the region of *PSMC3*. The three patients carried an homozygous deep intronic variation (i.e. > 100 bases from the exon–intron boundaries) (Vaz-Drago *et al*, 2017) with a predicted splicing effect on the *PSMC3* gene that we confirmed on the patient's cells. We then focused on demonstrating the effect of this variation at the level of the proteasome.

To our knowledge, this is the first report of a human biallelic pathogenic variant occurring in one of the ATPase Rpt subunits of the base of the 19S regulatory particle. Recently, *de novo* pathogenic variations in the non-ATPase subunit *PSMD12* (Rpn5) of the 19S regulator lid of the 26S complex have been reported in six patients with neurodevelopmental disorders including mainly intellectual disability (ID), congenital malformations, ophthalmic anomalies (no cataracts), feeding difficulties, deafness (unspecified type for two patients/6) and subtle facial features (Küry *et al*, 2017). *PSMD12* variants have been also associated with a large family with ID and autism and one simplex case with periventricular nodular heterotopia (Khalil *et al*, 2018). *PSMD12* is highly intolerant to loss-of-function (LoF) variations, and the most likely effect is haploinsufficiency due to the *de novo* heterozygote occurrence of loss-of-function truncating, non-sense or deletion variants. Interestingly, according to the gnomAD and DDD data (Huang *et al*, 2010; preprint: Karczewski *et al*, 2019), *PSMC3* is also predicted to be extremely intolerant to LoF variation. Indeed, the haploinsufficiency score of *PSMC3* is 4.76 that is within the high ranked genes (e.g. HI ranges from 0 to 10%) from the DECIPHER data. The pLI score (0.96) makes it among the highest intolerant genes (e.g. a score > 0.9 defines the highest range) with only three observed LoF variants vs. 23.2 predicted and confidence interval = 0.13). This explains also maybe the rarity of LoF variations found to date in this

gene. In our cases, this is the first time that biallelic class five variations are reported in one of the proteasome subunit delineating a recessive mode of inheritance. Nevertheless, the fact that the homozygous variation is affecting the splicing machinery and leads to a reduced but not abolished expression of *PSMC3* could mimic a possible haploinsufficiency mechanism although we cannot rule out a semi-dominant effect of the truncated Rpt5 form. It should also be noted that the acquisition of neurodevelopmental phenotypes upon proteasome dysfunction is not necessarily restricted to LoF variations in genes of the 19S regulatory particle, since recent work demonstrated that biallelic variants in the *PSMB1* subunit of the 20S core particle were associated with intellectual disability and developmental delay (Ansar *et al*, 2020).

Both proteomic and biochemical approaches undertaken in this study revealed that the deep intronic homozygous *PSMC3* variation is associated with increased amounts of 26S and 20S-PA28 proteasome complexes (Figs 2C and 3). The observation that patient cells concomitantly increase their intracellular pool of ubiquitin–protein conjugates (Figs 2A and 3D) is surprising and strongly suggests that such proteasomes are defective. In support of this notion, we found that, although carrying greater amounts of proteasomes, patient fibroblasts did not exhibit higher chymotrypsin-like activity compared to control cells, which can be at least partly explained by upregulation of PA28 and the concomitant increased peptide hydrolysis (Ma *et al*, 1992; Fig 3A and B). The C-terminus of Rpt5, which is supposed to be truncated in the patient due to the deep intronic splice variant, has been shown to be important for gate opening of the α-ring of the 20S proteasome core complex and for assembly of the 19S complex (Smith *et al*, 2007; Singh *et al*, 2014). Thus, an expression of this truncated Rpt5 variant even in low mounts may disturb proteasome assembly and function. These data led us to conclude that the increased steady-state expression level of the proteasome subunits observed in patient fibroblasts might reflect a constitutive *de novo* synthesis of proteasomes, which aims to compensate the dysfunctional ones. Strikingly and in contrast to control cells, TCF11/Nrf1 is constitutively processed in patient cells (Fig 4), confirming that patients' proteasomes were impaired. This, in turn, gives rise to a pathological vicious circle of events in which TCF11/Nrf1 and defective proteasomes stimulate each other (Fig 7). We reasoned that such a process may result in a pathway overload, which in turn reduces the ability of TCF11/Nrf1 to respond to further proteotoxic stress. Consistent with this hypothesis and unlike control cells, patient fibroblasts were not capable of upregulating their proteasome subunits when challenged with proteasome inhibitor carfilzomib (Fig 4). Importantly, overexpression of wild-type *PSMC3* could successfully rescue the phenotype of these cells by (i) restoring ubiquitin homeostasis (Fig 5A and B), (ii) sparing the TCF11/Nrf1 pathway (Fig 5C) and (iii) preserving proteasome subunit expression following carfilzomib treatment (Fig 5C). This result is of great importance, as it confirms that the deep intronic homozygous *PSMC3* variation in patient cells is responsible for their inability to cope with proteotoxic stress. Because cataract and semicircular canal malformations are observed in zebrafish embryos depleted with *PSMC3*, and a fortiori proteasomes, these data established a clear cause and effect relationship between the deep intronic *PSMC3* variant and the acquisition of patients' phenotype. On the other hand, one cannot exclude that the pathogenesis of the homozygous *PSMC3* variation may involve additional mechanisms.

Because target genes of TCF11/Nrf1 include anti-inflammatory factors (Widenmaier *et al*, 2017; Yang *et al*, 2018), it is also conceivable that inflammation might play a role in this process. This assumption would be in line with the observation that subjects suffering from other loss-of-function proteasome variations such as *PSMB8* (Agarwal *et al*, 2010; Arima *et al*, 2011; Liu *et al*, 2012), *PSMA3, PSMB4, PSMB9* (Brehm *et al*, 2015) and/or *POMP* (Poli *et al*, 2018) exhibit an inflammatory phenotype including joint contractures. This notion is further reinforced by the fact that patient fibroblasts constitutively express type I and/or II interferon (IFN) genes such as those encoding the immunoproteasome subunits β5i, β1i as well as PA28-α (Fig 4). In any case, the potential contribution of innate immunity to the pathogenesis of the homozygous *PSMC3* variant via TCF11/Nrf1 warrants further investigation (Sotzny *et al*, 2016; Poli *et al*, 2018).

The ubiquitin–proteasome system (UPS) is a protein degradation pathway that regulates the intracellular level of proteins involved in a very wide variety of eukaryote cellular functions. Thus, it is not surprising that this pathway is related to multiple human conditions including cataract, where an overburden of dysfunctional and aggregated proteins cannot be adequately removed by the UPS (Shang & Taylor, 2012). Moreover, protein degradation dysfunction is recognized as a widespread cause of neurodegenerative diseases such as Parkinson, Huntington or Alzheimer diseases. Several inherited rare disorders have been shown to be related to directly UPS dysfunction in enzymes of the ubiquitin conjugation machinery: *UBE3A* in Angelman syndrome (Kishino *et al*, 1997, p. 3), *UBE2A* in X-linked syndromic ID (Nascimento *et al*, 2006), *UBE3B* in Kaufman oculocerebrofacial syndrome (classified as blepharophimosis—mental retardation syndrome) or other related enzymes such as *HUWE1* an ubiquitin ligase in X-linked for dominant ID syndrome (Froyen *et al*, 2008; Moortgat *et al*, 2018) and *OTUD6B* a deubiquitinating enzyme for neurodevelopmental disability with seizures and dysmorphic features (Santiago-Sim *et al*, 2017). The phenotypic tropism for

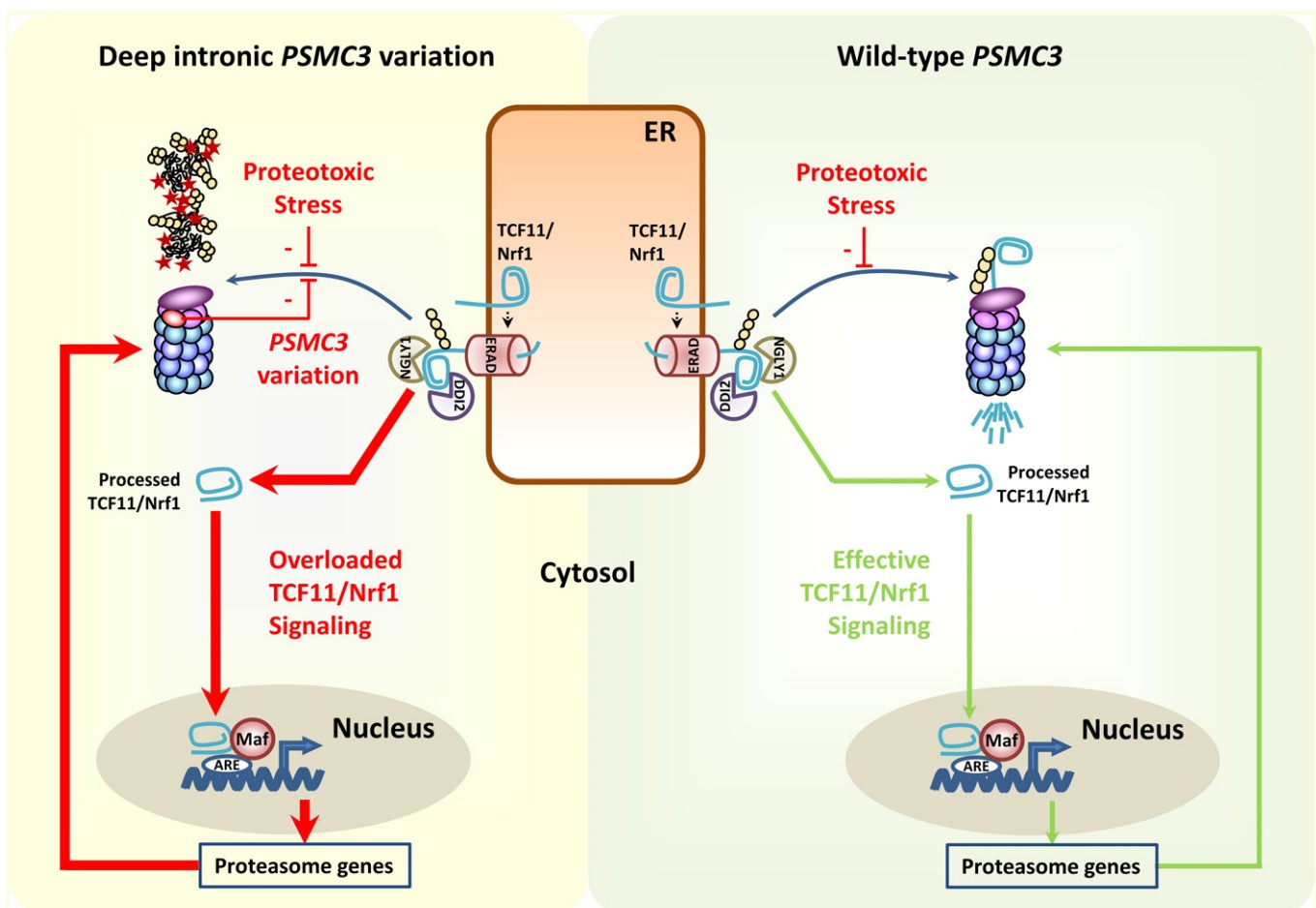

**Figure 7. Schematic diagram depicting TCF11/Nrf1 processing pathway in response to proteasome dysfunction and/or proteotoxic stress in patient carrying the deep intronic homozygous *PSMC3* variation (left) and in healthy subjects (right).**

Under normal conditions, TCF11/Nrf1 is a short-lived ER membrane (endoplasmic reticulum)-resident protein, which is rapidly subjected to proteasome-mediated degradation following retro-translocation to the cytosol by ER-associated degradation machinery (ERAD). In case of proteasome dysfunction (i.e. proteotoxic stress), the half-life of TCF11/Nrf1 is prolonged and become then a substrate for the NGLY1 and DDI2, thereby giving rise to a C-terminal cleaved fragment that enters into the nucleus. After nuclear translocation, cleaved TCF11/Nrf1 associates with Maf and promotes the expression of proteasomes genes, so that protein homeostasis can be restored. In patients carrying the c.1127 + 337A>G homozygous *PSMC3* variation, defective proteasomes promote a constitutive activation of TCF11/Nrf1, thereby resulting in increased assembly of newly synthetized non-functional proteasomes, which in turn activate TCF11/Nrf1 again. Over-activation of TCF11/Nrf1 results in pathway exhaustion thus rendering patient cells incapable of responding to further proteotoxic stress.

neurodevelopmental disorders especially with intellectual disability and other brain dysfunctions (autism, seizures, brain malformations) seems to be overall a hallmark of the gene alterations related to the UPS inherited dysfunctions. In the family presented herein, all three patients had various neurodevelopmental anomalies (autism, ataxia, mild ID) and mild facial dysmorphism.

However, the striking clinical presentation of the cases reported herein is the combination of early-onset deafness and early-onset cataracts that motivated (independently) the referral of the three cases. These features have not yet been related to this group of disorders to date. Moreover, this association has only been reported twice in larger syndromic forms such as the dominant form of *WFS1* but in the context of neonatal/infancy-onset diabetes (De Franco *et al*, 2017) and the Aymé-Gripp syndrome (including also intellectual disability, seizures and Down syndrome like facies) with *de novo* pathogenic variants in the *MAF* gene (Niceta *et al*, 2015). Variations in the later one were shown to impair *in vitro* MAF phosphorylation, ubiquitination and proteasomal degradation.

The zebrafish model was extremely useful to demonstrate the effect of a reduced *psmc3* expression leading to inner ear anomalies and lens opacities. The overall development of the lens and inner ear as well as the underlying gene regulatory networks are highly conserved throughout evolution (Cvekl & Zhang, 2017). Pathogenic variations or knockdown of genes implicated in human cataract and/or deafness in zebrafish has been shown to recapitulate similar phenotypes (Busch-Nentwich, 2004; Takamiya *et al*, 2016; Gao *et al*, 2017; Mishra *et al*, 2018; Yousaf *et al*, 2018). Using two strategies (morpholino and CRISPR/Cas9), the fish developed a cataract and ear phenotype that could be rescued. We did not observe other obvious anomalies. Interestingly, a reduction of proteasome activity has previously been associated with lens defects in zebrafish. The knockout of the zebrafish gene *psmd6* and the knock down of *psmd6* and *psmc2*, both encoding for proteins of the proteasome, resulted in a severe impairment of lens fibre development. Cataract was also proposed as a consequence of disrupted lens fibre differentiation (Richardson *et al*, 2017). The ear phenotype of the *psmd6* mutant and both psmd6 morphants was not assessed (Imai *et al*, 2010). A direct link between the UPS and auditory hair cell death or impaired semicircular canal morphogenesis has not been described in zebrafish yet. However, knockdown of *atoh1,* a gene regulated by the UPS, has been shown to severely affect hair cell development in the inner ear of zebrafish (Millimaki *et al*, 2007). The malformation of canal pillars observed in zebrafish *psmc3* morphants and crispants might be also a secondary effect, as abnormal sensory cristae with few hair cells have been previously assumed to lead to an abnormal development of semicircular canals (Haddon & Lewis, 1991; Cruz *et al*, 2009). Patients' hearing loss seems rapidly progressive—in the earlier months of life—rather than congenital: OAE were present at birth but had disappeared upon deafness diagnostic. Cochlear implant was possible in all three patients and the results are correlated with the patients' condition (autistic features) and time of implantation. Indeed, patients II.4 and II.7 show severe autistic features and were implanted late (i.e. after the age of 2 and more than a year after deafness diagnostic). They did not develop language and do not use their cochlear implant, and audiometric assessment is impossible. Patient II.2 was implanted at the age of 22 months (i.e. 7 months after deafness diagnostic), and his tone audiometry results are

excellent. Despite wearing his implant every day for 10 years, he still shows severe language delay.

In conclusion, our work demonstrates the implication of a deep intronic variant in a novel ultra-rare neurosensorial syndrome with early-onset cataract and deafness in one of the proteasome subunit, *PSMC3*. Although *de novo* dominant variations have been associated with several proteasome-related disorders, we report for the first time a biallelic pathogenic variant. Our observations strongly suggest that the amount of PSMC3 is critically implied in the development and maintenance of the inner ear and the lens.

# Materials and Methods

## Patients and ethics

The three cases have consulted independently and were enrolled subsequently by the CARGO (reference centre for rare eye diseases at the Strasbourg University Hospital, France). All participants were assessed by a clinical geneticist, a neuropaediatrician, an Ear-Nose-Throat specialist, a dermatologist and a paediatric ophthalmologist. Written consent for research and publication (including photography) was obtained for all study participants. This research followed the tenets of the Declaration of Helsinki. Approval was obtained from our institutional review board *"Comité Protection des Personnes"* (EST IV, N∘DC-20142222). The identified gene was submitted to the GeneMatcher tool, but no other patient with the same phenotype could be identified (Sobreira *et al*, 2015).

## Homozygosity regions

Three affected individuals (II.2, II.4 and II.7) and two unaffected individuals (II.1 and II.3) were analysed with the Affymetrix GeneChip® Mapping 250K Array Xba 240 (Affymetrix, Santa Clara, CA). Sample processing and labelling were performed according to the manufacturer's instructions. Arrays were hybridized on a Gene-Chip Hybridization Oven 640, washed with the GeneChip Fluidics Station 450 and scanned with a GeneChip Scanner 3000. Data were processed by the GeneChip DNA Analysis Software version 3.0.2 (GDAS) to generate SNP allele calls. An average call rate > 99% was obtained. Homozygosity regions were identified as regions of homozygosity longer than 25 adjacent SNPs. Shared regions of homozygosity are visualized by the HomoSNP software (IGBMC, Strasbourg), which displays one patient per line (Appendix Fig S1). Three regions of homozygosity of, respectively, 0.3, 2.1 and 1.3 Mb on chromosome 11 (11:44,396,024–44,668,374; 11:45,574,574–47,684,908; 11:66,066,993–67,349,899) are shared between the affected and not the healthy individuals.

## Whole-exome sequencing

Whole-exome sequencing (WES) was performed in 2014 for the three affected siblings (II.2, II.4, II.7) and one healthy brother (II.6) by the IGBMC (Institut de Génétique et de Biologie Moléculaire et Cellulaire, Illkirch-Graffenstaden, France) Microarray and Sequencing platform. Exons of DNA samples were captured using the in-solution enrichment methodology (Agilent SureSelect All Exon XT2 50 Mb Kit) and sequenced with an Illumina HiSeq 2500 instrument

to generate 100 bp paired-end reads. The reads were mapped to the reference human genome (GRCh37/hg19) using the Burrows-Wheeler Aligner (BWA v0.7.1) (Li & Durbin, 2009). The HaplotypeCaller module of the Genome Analysis ToolKit (GATK v3.4-46) was used for calling both SNV and indel (DePristo *et al*, 2011). Structural variations (SV) were called using CANOES (Backenroth *et al*, 2014).

From 96,222 to 102,072 genetic variants (SNV/indel/SV) were identified per individual from the WES analysis (Appendix Table S1). Bioinformatics analyses (see Materials and Methods) highlighted a unique homozygous missense variation in the *DGKZ* gene (NM_001105540.1:c.1834G>A, p.Ala612Thr) encoding for the Diacylglycerol Kinase Zeta, located in a homozygous region of interest on chromosome 11 (Appendix Table S1). This variant, reported previously with a gnomAD frequency of 0.018%, was predicted tolerated by SIFT (Ng & Henikoff, 2003) and neutral by PolyPhen-2 (Adzhubei *et al*, 2010) and was finally manually ruled out, as we were unable to explain the patients phenotype based on the gene function.

## Whole-genome sequencing

Whole-genome sequencing was performed in 2016 for the three affected siblings (II.2, II.4, II.7) and two healthy brothers (II.1, II.6) by the *Centre National de Recherche sur le Génome Humain* (CNRGH, Evry France). Genomic DNA was used to prepare a library for WGS using the Illumina TruSeq DNA PCR-Free Library Preparation Kit. After normalization and quality control, qualified libraries were sequenced on a HiSeq2000 platform (Illumina Inc., CA, USA), as paired-end 100 bp reads. At least three lanes of HiSeq2000 flow cell were produced for each sample, in order to reach an average sequencing depth of 30×. Sequence quality parameters were assessed throughout the sequencing run, and standard bioinformatics analysis of sequencing data was based on the Illumina pipeline to generate FASTQ files for each sample. The sequence reads were aligned to the reference sequence of the human genome (GRCh37) using the Burrows-Wheeler Aligner (BWA V7.12) (Li & Durbin, 2009). The UnifiedGenotyper and HaplotypeCaller modules of the Genome Analysis ToolKit (GATK) (DePristo *et al*, 2011), Platypus (https://github.com/andyrimmer/Platypus) and Samtools (Li *et al*, 2009) were used for calling both SNV and indel. Structural Variations (SV) were called using SoftSV (Bartenhagen & Dugas, 2016). Moreover, each known cataract and deafness genes were visually inspected with IGV (Thorvaldsdóttir *et al*, 2013).

## Bioinformatics analysis

Annotation and ranking of SNVs/indels and structural variations were performed, respectively, by VaRank (Geoffroy *et al*, 2015) (in combination with Alamut Batch, Interactive Biosoftware, Rouen, France) and by AnnotSV (Geoffroy *et al*, 2018a). Variant effect on the nearest splice site was predicted using MaxEntScan (Yeo & Burge, 2004), NNSplice (Reese *et al*, 1997) and Splice Site Finder (Shapiro & Senapathy, 1987). Very stringent filtering criteria were applied to filter out non-pathogenic variants: (i) variants represented with an allele frequency of more than 1% in public variation databases including the 1000 Genomes (1000 Genomes Project Consortium *et al*, 2015), the gnomAD database (Lek *et al*, 2016),

the DGV (MacDonald *et al*, 2014) or our internal exome database, and (ii) variants in 5′ and 3′ UTR, downstream, upstream, intronic and synonymous locations without pathogenic prediction of local splice effect. The *PSMC3* nomenclature is based on the accession number NM_002804.4 from the RefSeq database (O'Leary *et al*, 2016). Genomic coordinates are defined according to GRCh37/hg19 assembly downloaded from the University of California Santa Cruz (UCSC) genome browser (Tyner *et al*, 2017).

## Sanger confirmation and segregation

The variant confirmation and the cosegregation analysis with the phenotype in the family member were performed by Sanger sequencing after PCR amplification of 50 ng of genomic DNA template. The primers were designed with Primer 3 (http://frodo.wi.mit.edu/primer3) and are detailed in Appendix Table S6. Bidirectional sequencing of the purified PCR products was performed by GATC Sequencing Facilities (Konstanz, Germany).

## Plasmid construction

The DNA sequence for PSMC3 encoding the full-length Rpt5 proteasome subunit (NM_002804.4) was amplified by PCR from a pcDNA/Zeo (+) expression vector encoding a N-terminally HA-tagged Rpt5 (kind gift from Dr. Richard Golnik, Charité Universitätsmedizin Berlin) and cloned into pcDNA3.1/myc-HIS version B (Invitrogen) to construct an untagged PSMC3/Rpt5 using the EcoRV and XhoI restriction sites. The resulting pcDNA3.1/PSMC3 construct was sequenced (Microsynth Seqlab, Göttingen) before being used for transfection experiments.

## RNA and protein analysis using the patient's cells

Fibroblasts of patient II.4 and control individuals were obtained by skin biopsy as previously described (Scheidecker *et al*, 2015). Three sex and age matched controls were used.

RNA was extracted from skin fibroblasts of individual II.4 and a healthy unrelated control using RNeasy RNA kit (Qiagen); then, we performed reverse transcription using the iScriptTM cDNA Synthesis Kit (Bio-Rad, Hercules, CA).

Protein analysis includes Western blot for which extracted proteins using the RIPA Buffer (89901 Thermo Scientific) complemented with protease inhibitor cocktail (Roche 06538282001) from primary fibroblasts of affected and control individuals were loaded onto Mini-PROTEAN TGX gels (Bio-Rad).

Immunofluorescence assays were performed using primary fibroblasts from patient and control individuals grown in Nunc Lab-Tek chamber slides (Thermo Scientific, Waltham, MA, USA) fixed with 4% paraformaldehyde, incubated with PBS 0.5% Triton X-100 for 10 min and blocked with PBS-20% FCS. Cells were then incubated for 1 h with primary antibodies, washed three times in PBS, incubated for 1 h with secondary antibody and DAPI, washed again in PBS and mounted in Elvanol No-Fade mounting medium before observation on a fluorescence microscope (Zeiss Axio Observer D1) at a ×400 magnification. The primary antibodies were directed against the N- and C-terminal part of PSMC3. The secondary antibody was goat anti-mouse Alexa Fluor coupled 568 IgG (Invitrogen).

Accumulation of ubiquitinated proteins has been observed using a dedicated Western blot. Skin fibroblasts from three controls and the affected individual were recovered in ice-cold RIPA buffer with protease inhibitors ("Complete EDTA-free"; Roche Diagnostics, 1 tablet in 10 ml buffer) and 25 mM N-ethylmaleimide (NEM, diluted freshly in ethanol to prevent artefactual deubiquitination), left on ice for 45 min, centrifuged for 10 min at 13,360 *g*, supernatant recovered and 5× Laemmli buffer added. Bands were quantified relative to the total amount of protein loaded (stainfree) using Image Lab. All bands corresponding to the various ubiquitinated proteins were added and expressed relative to the amount of ubiquitinated proteins in control 1 fibroblasts. The mean of six independent experiments was calculated and represented as a histogram, and a Student's *t*-test was performed to determine the significance. Primary and secondary antibodies used in these experiments study as well as their dilution are described in Appendix Table S7.

## Co-immunoprecipitation and mass spectrometry analysis

PSMC3 protein and its partners were immunoprecipitated from patient's fibroblasts using magnetic microparticles (MACS purification system, Miltenyi Biotec) according to the manufacturer's instructions and as previously described (Stoetzel *et al*, 2016, 15). Briefly, PSMC3 complexes were captured with an anti-PSMC3 antibody (Abcam ab171969) and the target and its associated proteins were purified on the protein G microbeads (Miltenyi Biotec). Proteins were eluted out of the magnetic stand with 1× Laemmli buffer (Bio-Rad). To optimize reproducibility, co-immunoprecipitation experiments were carried out in affinity triplicates on exactly 1.1 mg of proteins for each sample. For negative controls, halves of each sample were pooled and immunoprecipitated with the protein G beads, omitting the antibody.

Protein extracts were prepared as described in a previous study (Waltz *et al*, 2019). Each sample was precipitated with 0.1 M ammonium acetate in 100% methanol, and proteins were resuspended in 50 mM ammonium bicarbonate. After a reduction–alkylation step (dithiothreitol 5 mM–iodoacetamide 10 mM), proteins were digested overnight with sequencing-grade porcine trypsin (1:25, w/w, Promega, Fitchburg, MA, USA). The resulting vacuum-dried peptides were resuspended in water containing 0.1% (v/v) formic acid (solvent A). One fifth of the peptide mixtures were analysed by nanoLC-MS/MS an Easy-nanoLC-1000 system coupled to a Q-Exactive Plus mass spectrometer (Thermo Fisher Scientific, USA) operating in positive mode. Five microlitres of each sample was loaded on a C-18 precolumn (75 μm ID × 20 mm nanoViper, 3 μm Acclaim PepMap; Thermo) coupled with the analytical C18 analytical column (75 μm ID × 25 cm nanoViper, 3 μm Acclaim PepMap; Thermo). Peptides were eluted with a 160-min gradient of 0.1% formic acid in acetonitrile at 300 nl/min. The Q-Exactive Plus was operated in data-dependent acquisition mode (DDA) with Xcalibur software (Thermo Fisher Scientific). Survey MS scans were acquired at a resolution of 70K at 200 m/z (mass range 350–1,250), with a maximum injection time of 20 ms and an automatic gain control (AGC) set to 3e6. Up to 10 of the most intense multiply charged ions ($\geq$ 2) were selected for fragmentation with a maximum injection time of 100 ms, an AGC set at 1e5 and a resolution of 17.5K. A dynamic exclusion time of 20 s was applied during the peak selection process.

MS data were searched against the Swiss-Prot database (release 2019_05) with Human taxonomy. We used the Mascot algorithm (version 2.5, Matrix Science) to perform the database search with a decoy strategy and search parameters as follows: carbamidomethylation of cysteine, N-terminal protein acetylation and oxidation of methionine were set as variable modifications; tryptic specificity with up to three missed cleavages was used. The mass tolerances in MS and MS/MS were set to 10 ppm and 0.05 Da, respectively, and the instrument configuration was specified as "ESI-Trap". The resulting ".dat" Mascot files were then imported into Proline v1.4 package (http://proline.profiproteomics.fr/) for post-processing. Proteins were validated with Mascot pretty rank equal to 1, 1% FDR on both peptide spectrum matches (PSM) and protein sets (based on score). The total number of MS/MS fragmentation spectra was used to quantify each protein in the different samples.

For the statistical analysis of the co-immunoprecipitation data, we compared the data collected from multiple experiments against the negative control IPs using a homebrewed R package as described previously (Chicois *et al*, 2018) except that that the size factor used to scale samples was calculated according to the DESeq normalization method [i.e. median of ratios method (Anders & Huber, 2010)]. The package calculates the fold change and an adjusted *P*-value corrected by Benjamini–Hochberg for each identified protein (and visualizes the data in volcano plots).

## Native-PAGE and proteasome in-Gel peptidase activity assay

Patient and control fibroblasts were lysed in equal amounts of TSDG buffer (10 mM Tris pH 7.0, 10 mM NaCl, 25 mM KCl, 1.1 mM MgCl$_2$, 0.1 mM EDTA, 2 mM DTT, 2 mM ATP, 1 mM NaN$_3$, 20% Glycerol) prior to protein quantification using a standard BCA assay (Pierce) following the manufacturer's instructions. Twenty micrograms of whole-cell extracts were loaded on 3–12% gradient native-PAGE gels (Invitrogen) and subsequently subjected to a 16-h electrophoresis at 45 V using a running buffer consisting of 50 mM BisTris/Tricine, pH 6.8. Chymotrypsin-like activity of the separated proteasome complexes was then measured by incubating 0.1 mM of the suc-LLVY-AMC substrate (Bachem) at 37°C for 20 min in a final volume of 10 ml of overlay buffer (20 mM Tris, 5 mM MgCl$_2$, pH 7.0). Proteasome bands were then visualized by exposure of the gel to UV light at 360 nm and detected at 460 nm using an Imager.

## Measurement of proteasome activity in crude extracts

Ten micrograms of whole-cell extracts deriving from control and patient fibroblasts was tested for chymotrypsin-like activity by exposing the cell lysates to 0.1 mM of the Suc-LLVY-AMC peptide. Assays were carried out in a 100 μl reaction volume of ATP/DTT lysis buffer at 37°C. The rate of cleavage of fluorogenic peptide substrate was determined by monitoring the fluorescence of released aminomethyl-coumarin using a plate reader at an excitation wavelength of 360 nm and emission wavelength of 460 nm over a period of 120 min.

## Western blot analysis

For Western blotting, equal amounts of RIPA (50 mM Tris pH 7.5, 150 mM NaCl, 2 mM EDTA, 1 mM N-ethylmaleimide, 10 μM MG-132, 1% NP-40, 0.1% SDS) buffered protein extracts from control

and patient fibroblasts were separated in SDS-Laemmli gels (12.5 or 10%). Briefly, following separation, proteins were transferred to PVDF membranes (200 V for 1 h) blocked with 1× Roti®-Block (Carl Roth) for 20 min at room temperature under shaking and subsequently probed overnight at 4°C with relevant primary antibodies. The anti-Alpha6 (clone MCP20), anti-Beta1 (clone MCP421), anti-Beta2 (clone MCP165), anti-Rpt1 (PSMC2, BML-PW8315), anti-Rpt2 (PSMC1, BML-PW0530), anti-Rpt3 (PSMC4, clone TBP7-27), anti-Rpt4 (PSMC6, clone p42-23), anti-Rpt5 (PSMC3, BML-PW8310) and anti-Rpt6 (PSMC5, clone p45-110) primary antibodies were purchased from Enzo Life Sciences. The anti-Beta1i (LMP2, K221) and anti-PA28-α (K232/1) are laboratory stocks and were used in previous studies(Poli *et al*, 2018). Antibodies specific for Beta5 (ab3330) and α-tubulin (clone DM1A, ab7291) were purchased from Abcam. Primary antibodies specific for TCF11/Nrf1 (clone D5B10) and ubiquitin (clone D9D5) were obtained from Cell Signaling Technology. The anti-Beta5i antibody (LMP7, clone A12) was a purchase from Santa Cruz Biotechnology Inc. After incubation with the primary antibodies, membranes were washed three times with PBS/ 0.4% Tween and incubated with anti-mouse or anti-rabbit HRP conjugated secondary antibodies (1/5,000). Visualization of immunoreactive proteins was performed with enhanced chemiluminescence detection kit (ECL) (Bio-Rad). The ImageJ 1.48v software was used for densitometry analysis of the ECL signals.

## Zebrafish analysis

### Zebrafish (Danio rerio) maintenance and husbandry
In this study, the zebrafish wild-type line AB2O2 (University of Oregon, Eugene) and the transgenic line gSAIzGFF539A (National Institute of Genetics, Mishima, Japan) were used and maintained at 28°C under a 14-h light and 10-h dark cycle as described previously (Westerfield, 2000). When fish reached sexual maturity, zebrafish couples were transferred to breeding tanks the day before and crossed after the beginning of the next light cycle. Fertilized zebrafish eggs were raised at 28.5°C in 1× Instant Ocean salt solution (Aquarium Systems, Inc.). Zebrafish husbandry and experimental procedures were performed in accordance with German animal protection regulations (Regierungspräsidium, Karlsruhe, Germany, AZ35-9185.81/G-137/10). In all experiments, samples (*n*) represent a random selection of a bigger cohort. The status of the zebrafish was always known to the experimenters.

### Microinjections
Injections were performed as described before (Müller *et al*, 1999). Morpholinos *psmc3-mo* (TGTGAATCACAGTATGAAGCGTGCC, Genetools LLC, Oregon) and *ctrl-mo* (5-bp mismatched) (TGTCAATGAGAGTATCAACCGTGCC, Genetools LLC, Oregon) were injected at 0.2 mM (Appendix Fig S16). CRISPR guide RNAs were designed with ChopChop software (Labun *et al*, 2019) and synthesized with the MEGAshortscript T7 Transcription Kit (Ambion) according to the manufacturer's instructions. For CRISPR/Cas9 injections, 300 ng/μl of Cas9 protein (GeneArt Platinum Cas9 Nuclease, Invitrogen) and 300 ng/μl guide RNA were combined. Guide RNA1 (G1 binding sequence: AGATCCTAATGACCAAGAGGAGG) targets exon 4, and Guide RNA2 (G2 binding sequence: CAGGATATCCACCCTGTTAGTGG) targets exon 9. The Web interface PCR-F-Seq q (http://iai-gec-server.iai.kit.edu) was used to quantify the cutting efficiency of both

guide RNAs (Etard *et al*, 2017). For the life imaging experiment, the transgenic line gSAIzGFF539A, marking the semicircular canals, was generated using the Gal4-UAS system as described previously (Asakawa *et al*, 2008). For rescue experiments, mRNA of zebrafish *psmc3* was synthesized using the mMESSAGE mMACHINE system (Ambion). Zebrafish *psmc3* mRNA was injected at a final concentration of 10 ng/μl.

### PCR amplification and cloning
Total RNA from 24 to 72 hpf zebrafish embryos was extracted using Tri-reagent (Invitrogen, Carlsbad, CA) and reverse transcribed using M-MLV Reverse Transcriptase (Promega, Germany). To analyse the expression pattern of *psmc3* in zebrafish embryos, we amplified and cloned a 956 bp *psmc3* fragment into the pGEMT-easy vector (Promega). For the synthesis of DIG-labelled RNA probes, we used Apa1 to linearize the plasmid and SP6 to transcribe the anti-sense RNA DIG probes. To exclude possible off-target effects caused by morpholino or CRISPR/Cas9 injection, a rescue experiment was performed. Full-length *psmc3* was cloned into pCS2+ using EcoRI, linearized with NotI and *psmc3* full-length RNA synthesized using the mMESSAGE mMACHINE SP6 Kit (Ambion). To verify the efficiency of morpholino (*psmc3-mo*), we extracted total RNA from morpholino-injected embryos using Tri-reagent (Invitrogen, Carlsbad, CA), transcribed mRNA into cDNA using M-MLV Reverse Transcriptase (Promega, Germany) and checked the effect on the *psmc3* splice sites by RT–PCR using the amplification conditions as described before (Appendix Fig S16B; Müller *et al*, 1999). Using Sanger Sequencing, the CRISPR/Cas9 efficiency was assessed (Appendix Fig S16C and D).

### Whole-mount in situ hybridization
*In situ* hybridization was performed as previously described (Crow & Stockdale, 1986; Oxtoby & Jowett, 1993; Costa *et al*, 2002). To suppress melanogenesis, 20 hpf zebrafish embryos were transferred and raised in water supplemented with 200 mM 1-phenyl 2-thiourea (PTU). Probes targeting the messenger RNAs of the genes *krox20*, *msxc*, *her8a* and *sox19b* were obtained from Armant *et al* (2013). Plasmids to generate probes targeting *otopetrin*, *versican* and *versican b* were provided by the laboratory of Tanya Whitfield.

### Immunohistochemistry
Immunostaining of whole zebrafish embryos after 5 dpf was performed as previously described (Leventea *et al*, 2016) and imaged with Leica TCS SP5 confocal microscope. Antibodies are indicated in Appendix Table S7. For TUNEL staining, we used the ApopTag® Red In Situ Apoptosis Detection Kit (Millipore). For imaging the hair cells, we incubated 5 dpf embryos in a solution of 6 μM of FM1-43 (Life technology), for 1 min, wash them in E3 water, anaesthetized with 0.0168% (w/v) MESAB and embedded them into 0.5% (w/v) low melting point agarose chilled to 37°C in a lateral position.

### Microscopy
To examine the eye and ear phenotype in morphants and crispants, we anaesthetized living embryos with 0.0168% (w/v) MESAB (tricaine methanesulfonate, MS-222; Sigma-Aldrich, Taufkirchen, Germany) and embedded them into 0.5% (w/v) low melting point agarose chilled to 37°C in a lateral position with one eye/ear facing towards the objective. Confocal reflection microscopy was used to

examine zebrafish eyes for abnormal light reflection evoked through cataract as previously described (Takamiya *et al*, 2016) and imaged with the Leica TCS SP2 confocal system with a 63× water immersion objective. Brightfield and fluorescence real-time imaging of zebrafish ears were acquired using the Leica TCS SP5 confocal microscope with a 63× and 40× water immersion objective.

## Statistics

For *in vitro* and *in vivo* experiments, sample sizes were chosen according to the standard practice in the field. For humans, sample size was limited by the patient's sample available. In all zebrafish experiments, samples (*n*) represent a random selection of a bigger cohort. The status of the zebrafish was always known to the experimenters. Results are reported as mean ± standard deviation (SD) or (SEM) for the number of experiments indicated in the legends of each figure or table. Statistical analyses were performed using either Excel (Microsoft, USA) or the Prism software (GraphPad, USA). In most of the cases, we used a Student's *t*-test to compare two groups of normally distributed data. Significance levels and number of samples/replicates are indicated in each figure legends. All *P*-values for main Figures and appendix figures can be found in the Appendix Table S8.

# Data availability

The data sets produced in this study are available in the following databases:

- WGS sequences: European Genome-Phenome Archive (EGA) EGAS00001003942 (https://ega-archive.org/studies/EGAS00001003942)

- Protein interaction AP-MS data: Protein identifications database (PRIDE) PXD015836 (http://www.ebi.ac.uk/pride/archive/projects/PXD015836)

- Human variations: ClinVar SCV000864220.1 (https://www.ncbi.nlm.nih.gov/clinvar/SCV000864220)

Expanded View for this article is available online.

## The paper explained

### Problem
Early-onset deafness is one of the most common causes of developmental disorder in children (prevalence of 2–4/1,000 infants), and similarly, early-onset cataract is the most important cause of paediatric visual impairment worldwide (prevalence 2–13.6/10,000) accounting for 10% of the causes of childhood blindness. Many genes have already been implicated for each of these conditions independently but only very few genes identified when cataract and early-onset deafness occur together. Indeed, patients with both conditions are very rare and often linked to teratogenic exposure during pregnancy. In this study, we therefore aim at identifying the molecular cause of cases with syndromic early-onset deafness and congenital cataracts.

### Results
Using whole-genome sequencing combined with homozygosity mapping and bioinformatics prioritization, we uncovered a homozygous deep intronic variation in the *PSMC3* gene in three affected patients from a very large consanguineous family presenting with early-onset cataract and deafness. *PSMC3* encodes one of the proteasome subunits and more specifically Rpt5, the 26S proteasome AAA-ATPase subunit of the 19S proteasome complex responsible for recognition, unfolding and translocation of substrates into the 20S proteolytic cavity of the proteasome. The ubiquitin–proteasome system (UPS) is a major cellular system that degrades ubiquitin-modified proteins to maintain protein homeostasis and to control signalling including development. Using multiple approaches including proteomics, cellular biology (on patient's cells) and animal modelling (zebrafish), we could confirm the impact of this variation (transcription of a cryptic exon introducing a stop codon) on proteasome function and linked this to the patients' phenotype.

### Impact
This is the first implication of the 26S proteasome AAA-ATPase subunit in a disease with the description of biallelic pathogenic variations in the context of the emerging proteasomopathies that include *PSMA3, PSMB1, PSMB4, PSMB8, PSMB9, PSMD12, PSMG2* and *POMP* genes that are linked to loss-of-function variations. These results open new insights into the role of the proteasome or at least *PSMC3* in inner ear, lens and central nervous system development. Identifying novel genes implicated in such rare diseases will improve the genetic counselling and give affected families the opportunity to have access to preimplantation genetic diagnosis and prenatal testing.

## Acknowledgements

We would like to thank the patients and their family for their participation. Whole-exome sequencing and SNP arrays was performed by the IGBMC Microarray and Sequencing platform, a member of the "France Génomique" consortium (ANR-10-INBS-0009) and funded thanks to the support of the Centre Régional de Génétique Médicale de Strasbourg (CREGEMES). Whole-genome sequencing was supported by the Laboratory of Excellence GENMED (Medical Genomics) grant no. ANR-10-LABX-0013 managed by the National Research Agency (ANR) part of the Investment for the Future (*Investissement d'Avenir*) programme. This work was also supported by grants from the Fritz-Thyssen Foundation (Az. 10.16.2.022MN), the German Research Foundation (DFG SFBTRR186 A13 and SFBTRR 167 A04) to EK, the Molecular Medicine Research Consortium of the University of Greifswald (FOVB-2018-11 to FE), the National BioResource Project (NBRP), the NBRP and NBRP/Fundamental Technologies Upgrading Program from AMED to KK, the JED-Belgique foundation to JM/HD and the NBRP Fundamental Technologies Upgrading Program from AMED

(Japan Agency for Medical Research and Development). A.K-H. is supported by a doctoral fellowship from the "Initiatives d'Excellence" (IdEx) through the University of Strasbourg and by the Franco-German University (UFA/DFH). S.F. and SB are supported by CNRS/Université de Strasbourg and S.B. also by INSERM. We wish to thank our colleagues around the world for testing their cohorts of patients (Elena Semina, Patrick Calvas, Sandrine Marlin, Alain Verloes). We thank Frederic Plewniak for the use of HomoSNP. We thank Tanja Whitfield for sharing plasmids for the examination of the zebrafish ear. The computing resources for this work were provided by the BICS and BISTRO bioinformatics platforms in Strasbourg. The mass spectrometry instrumentation was funded by the University of Strasbourg, IdEx "Equipement mi-lourd" 2015.

## Author contributions

ES, SS, FSu, VP, CE, CS-S, VL, DL and HD gathered data from patients and performed clinical investigations; SM, AB, FSa, J-FD and VG gathered sequencing

data and performed analyses; AK-H, FE, CS, SB, BAZ and SF performed cell biology experiments and data analyses; CS, LK, JC and PH designed and performed the mass spectrometry experiments and data analyses; AK-H, MT, KK, CE and US designed and performed the zebrafish experiments and data analyses; AK-H, JM, US and HD analysed the data and wrote the paper; FE, VG, ES, SS, SF, SB, CS, US, PH, CE and EK contributed to manuscript writing; EK, JM, US and HD provided direction for the project, conceived and designed the experiments.

## Conflict of interest

The authors declare that they have no conflict of interest.

## For more information

The URLs for data presented in this article are as follows:

(i)    1,000 genomes, http://www.1000genomes.org
(ii)   Alamut, https://www.interactive-biosoftware.com/
(iii)  AnnotSV, https://www.lbgi.fr/AnnotSV
(iv)   ChopChop, https://chopchop.cbu.uib.no
(v)    Clinvar, https://www.ncbi.nlm.nih.gov/clinvar
(vi)   dbSNP, http://www.ncbi.nlm.nih.gov/projects/SNP
(vii)  Ensembl, http://www.ensembl.org
(viii) European Genome-Phenome Archive (EGA), https://www.ebi.ac.uk/ega
(ix)   GATK, https://gatk.broadinstitute.org
(x)    gnomAD database, https://gnomad.broadinstitute.org
(xi)   Human genome build (GRCh37/hg19), https://www.ncbi.nlm.nih.gov/grc
(xii)  Human Splice Finder, http://www.umd.be/HSF3
(xiii) NNSplice, http://www.fruitfly.org/seq_tools/splice.html
(xiv)  Online Mendelian Inheritance in Man (OMIM), http://www.omim.org
(xv)   PolyPhen-2, http://genetics.bwh.harvard.edu/pph2
(xvi)  Primer 3, http://frodo.wi.mit.edu/primer3
(xvii) PRoteomics IDEntifications database (PRIDE), https://www.ebi.ac.uk/pride
(xviii) RefSeq, https://www.ncbi.nlm.nih.gov/refseq
(xix)  SIFT, http://sift.jcvi.org
(xx)   UCSC Genome Bioinformatics, http://genome.ucsc.edu
(xxi)  UniProt, http://www.uniprot.org
(xxii) VaRank, http://www.lbgi.fr/VaRank

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
