## [Review Process File · EMBO Molecular Medicine]

PSMC3 variants cause neurosensory syndrome combining deafness and cataract due to proteotoxic stress

Ariane Kroell Hermi, Frederic Ebstein, Corinne Stoetzel, Veronique Geoffroy, Elise Schaefer, Sophie Scheidecker, Severine Bar, Masanari Takamiya, Koichi Kawakami, Barbara Zieba, Fouzia Studer, Valerie Pelletier, Carine Eyermann, Claude Speeg-Schatz, Vincent Laugel, Dan Lipsker, Florian Sandron, Steven McGinn, Anne Boland, Jean-François DELEUZE, Lauriane Kuhn, Johana Chicher, Philippe Hammann, Sylvie Friant, Christelle Etard, Elke Kruger, Jean Muller, Uwe Strähle, and Helene Dollfus

DOI: 10.15252/emmm.201911861

Corresponding author(s): Jean Muller (jeanmuller@unistra.fr), Uwe Strähle (uwe.straehle@kit.edu), Elke Kruger (elke.krueger@uni-greifswald.de), Helene Dollfus (dollfus@unistra.fr)

Review Timeline:

Submission Date:	5th Dec 19
Editorial Decision:	10th Jan 20
Revision Received:	31st Mar 20
Editorial Decision:	15th Apr 20
Revision Received:	4th May 20
Accepted:	7th May 20

Editor: Celine Carret

Transaction Report:

9th Jan 2020

Dear Dr. Muller,

Thank you for the submission of your manuscript to EMBO Molecular Medicine. We have now heard back from the two referees whom we asked to evaluate your manuscript.

You will see from the set of reports pasted below that both referees are enthusiastic about the paper while still requesting additional supporting experiments to be performed along with some clarifications and details. Of interest, both referees request a rescue experiment, ref. #1 on the fish model, ref. #2 on the cellular model. We would leave it to you whether to perform the rescue experiment in cells or in the fish model. Further, ref. #1 would like to see a more thorough characterisation of the deafness phenotype of patients and ref. #2 would like data to be shown from more than one patient. We believe that following this line of revision would improve the clinical relevance of the findings as well as strengthen its conclusiveness.

We would therefore welcome the submission of a revised version within three months for further consideration and would like to encourage you to address all the criticisms raised as suggested to improve conclusiveness and clarity. Please note that EMBO Molecular Medicine strongly supports a single round of revision and that, as acceptance or rejection of the manuscript will depend on another round of review, your responses should be as complete as possible.

I look forward to receiving your revised manuscript.

Yours sincerely,

Celine Carret

Celine Carret, PhD
Senior Editor
EMBO Molecular Medicine

*** Instructions to submit your revised manuscript ***

**** PLEASE NOTE **** As part of the EMBO Publications transparent editorial process initiative (see our Editorial at <https://www.embopress.org/doi/pdf/10.1002/emmm.201000094>), EMBO Molecular Medicine will publish online a Review Process File to accompany accepted manuscripts.

To submit your manuscript, please follow this link:

Link Not Available

- 1) a .doc formatted version of the manuscript text (including Figure legends and tables). Please make sure that the changes are highlighted to be clearly visible to referees and editors alike.
- 2) separate figure files*
- 3) supplemental information as Expanded View and/or Appendix. Please carefully check the authors guidelines for formatting Expanded view and Appendix figures and tables at <https://www.embopress.org/page/journal/17574684/authorguide#expandedview>
- 4) a letter INCLUDING the reviewers' reports and your detailed responses to their comments (as Word file)

Also, and to save some time should your paper be accepted, please read below for additional information regarding some features of our research articles:

- 5) The paper explained: EMBO Molecular Medicine articles are accompanied by a summary of the articles to emphasize the major findings in the paper and their medical implications for the non-specialist reader. Please provide a draft summary of your article highlighting
 - the medical issue you are addressing,
 - the results obtained and
 - their clinical impact.

- 6) For more information: There is space at the end of each article to list relevant web links for further consultation by our readers. Could you identify some relevant ones and provide such information as well? Some examples are patient associations, relevant databases,

OMIM/proteins/genes links, author's websites, etc...

7) Author contributions: the contribution of every author must be detailed in a separate section (before the acknowledgments).

8) EMBO Molecular Medicine now requires a complete author checklist (<https://www.embopress.org/page/journal/17574684/authorguide>) to be submitted with all revised manuscripts. Please use the checklist as a guideline for the sort of information we need WITHIN the manuscript as well as in the checklist. This is particularly important for animal reporting, antibody dilutions (missing) and exact p-values and n that should be indicated instead of a range.

9) Every published paper now includes a 'Synopsis' to further enhance discoverability. Synopses are displayed on the journal webpage and are freely accessible to all readers. They include a short stand first (maximum of 300 characters, including space) as well as 2-5 one sentence bullet points that summarise the paper. Please write the bullet points to summarise the key NEW findings. They should be designed to be complementary to the abstract - i.e. not repeat the same text. We encourage inclusion of key acronyms and quantitative information (maximum of 30 words / bullet point). Please use the passive voice. Please attach these in a separate file or send them by email, we will incorporate them accordingly.

You are also welcome to suggest a striking image or visual abstract to illustrate your article. If you do please provide a jpeg file 550 px-wide x 400-px high.

10) A Conflict of Interest statement should be provided in the main text

11) Please note that we now mandate that all corresponding authors list an ORCID digital identifier. This takes <90 seconds to complete. We encourage all authors to supply an ORCID identifier, which will be linked to their name for unambiguous name identification.

Currently, our records indicate that the ORCID for your account is 0000-0002-7682-559X.

Link Not Available

12) The system will prompt you to fill in your funding and payment information. This will allow Wiley to send you a quote for the article processing charge (APC) in case of acceptance. This quote takes into account any reduction or fee waivers that you may be eligible for. Authors do not need to pay any fees before their manuscript is accepted and transferred to our publisher.

Photos 400-800 DPI

Figures are not edited by the production team. All lettering should be the same size and style; figure

panels should be indicated by capital letters (A, B, C etc). Gridlines are not allowed except for log plots. Figures should be numbered in the order of their appearance in the text with Arabic numerals. Each Figure must have a separate legend and a caption is needed for each panel.

*Additional important information regarding figures and illustrations can be found at <http://bit.ly/EMBOPressFigurePreparationGuideline>

***** Reviewer's comments *****

Referee #1 (Comments on Novelty/Model System for Author):

The patients' cells were essential to confirm in the human genomic context the outcome of the pathogenic variant, and dissect its molecular and cellular effects. The zebrafish model enabled illustration in vivo of Psmc3 knockdown.

Referee #1 (Remarks for Author):

In this paper, upon WGS of 3 patients & 2 healthy controls, Kröll-Hermi et al report the first biallelic pathogenic variant occurring in one of the ATPase Rpt subunits of the base of the 19S regulatory particle in human. Using patients' cells, carrying the deep intronic homozygous PSMC3 pathogenic variant, the authors show that proteasome from patients while in greater number, are ineffective. Based on these data & other experiments performed in cellular and zebrafish PSMC3/Psmc3 models, the authors conclude that the origin of the observed phenotypes could be linked to proteasome deficit, due to an haploinsufficiency mechanism.

Substantial amounts of genetic, molecular and some functional data are reported supporting the key involvement of PSM3 mutation in a new syndrome, associating severe deafness and early-onset cataracts, plus some neurological and cutaneous symptoms. There are some issues regarding the correlations versus causality between the data in the PSMC3/Psmc3 models, and clinical features in patients that the authors could answer to improve further the manuscript.

Major comments:

1 - The expression profile of PSMC3 in the eye, ear, and brain is not clear. The authors refer to ubiquitous expression, but high magnification images in studied tissues showing cellular expression (epithelial, neuronal, support, mesenchymal cells ...) are necessary.

2- About zebrafish models, the authors describe the presence of 2 zebrafish psmc3 isoforms that share 83% sequence identity with the human orthologue. It was not clear if the MO used target one, or both isoforms.

Could the authors try a rescue using the human PSCM3 gene?

3- The link between proteasome abnormalities observed in patients' cells and the morphological abnormalities observed in zebrafish is missing?

4- Deafness is one major phenotype common to all 3 PSMC3-affected patients, It would be interesting to add a detailed phenotypic description (symmetry, age of onset, progression, &

severity of the hearing loss) of the hearing loss. The fact that patients have cochlear implant may indicate profound hearing loss, but the presentation of patient audiogram (& age) could highlight some genotype-phenotype correlations that would be great to discuss and will add more depth to the paper. For instance, do the patients present (or have presented) signs of balance problems? Are there CT scans of MRI data from patients that support defects in circular canal formation? The origin of deafness is not clear. In the paper, the authors have extensively characterized the defects in semicircular canal morphogenesis, which has nothing to do with hearing. More information on the morphology and function of the hair cells would be more informative. Besides showing reduction of kinocilia, what about hair cells (& support cells) morphology? hair bundle architecture? The authors could use FM1-43 that provide indication of hair cells activity.

Minor comments:

2- Patients present some variability in phenotype; what's the authors explanation for those differences?

4- What about cell death (TUNEL, caspase ...) in the eye and ear of zebrafish models?

3- It's not clear if haploinsufficiency is the sole mechanism taking place in patients. Heterozygotes (50% of protein) are normal. Patients cells show no difference in protein localization, but it's not clear how much wild-type protein (or transcript) is present despite the PSMC3 pathogenic variant. Is the truncated protein still present or not? If the truncated form might have some gain (or semi-dominant) function is not clear.

1- There are many typographical errors (missing spaces; most references positioned after the "."; ref TANAKA (not in capitals),...)

e.g. Page 9: last line:

... in the inner ear of zebrafish.(Millimaki et al., 2007) The of canal pillars observed in zebrafish psmc3

Referee #2 (Comments on Novelty/Model System for Author):

This is an interesting paper. If the authors follow my (easy) suggestions the technical quality would move to high. This is a new example of another proteasomal subunit that -when mutated- would cause neurological pathologies.

Referee #2 (Remarks for Author):

This is an interesting manuscript where the authors identified a mutation in the 19S proteasomal subunit PSMC3 that relates to malformation and a range of neurological disorders. They report that this mutation results in an accumulation of ubiquitinated species in patient fibroblasts, which is -as expected- not associated to the activity of the proteasome but likely the transfer into the proteasomal proteolytic cage for degradation. As a result, the patient fibroblasts do not really respond to proteotoxic stress. Importantly, the authors could mimic the patient's phenotypes in zebrafish where the PSMC3 was eliminated by morpholinos.

This is important work and reveals another proteasome related case for neurological diseases. I have only a few points.

1. The data are shown with fibroblasts from only one (out of three) patients and it would be better to show at least one other patient's fibroblast as well.

2. The experiments would be even better when normal psmc3 is overexpressed in the fibroblast to compete mutant psmc3 away and then show that the phenotypes (ubiquitin accumulation, responses to proteolytic stress) are corrected.
3. PA28 is highly expressed in the patient fibroblasts. What is the reason for this effect? And why does it decrease following proteotoxic stress?

Referee #1 (Comments on Novelty/Model System for Author):

The patients' cells were essential to confirm in the human genomic context the outcome of the pathogenic variant, and dissect its molecular and cellular effects. The zebrafish model enabled illustration *in vivo* of Psmc3 knockdown.

Referee #1 (Remarks for Author):

In this paper, upon WGS of 3 patients & 2 healthy controls, Kröll-Hermi et al report the first biallelic pathogenic variant occurring in one of the ATPase Rpt subunits of the base of the 19S regulatory particle in human. Using patients' cells, carrying the deep intronic homozygous PSMC3 pathogenic variant, the authors show that proteasome from patients while in greater number, are ineffective. Based on these data & other experiments performed in cellular and zebrafish PSMC3/Psmc3 models, the authors conclude that the origin of the observed phenotypes could be linked to proteasome deficit, due to an haploinsufficiency mechanism.

Substantial amounts of genetic, molecular and some functional data are reported supporting the key involvement of PSM3 mutation in a new syndrome, associating severe deafness and early-onset cataracts, plus some neurological and cutaneous symptoms.

There are some issues regarding the correlations versus causality between the data in the PSMC3/Psmc3 models, and clinical features in patients that the authors could answer to improve further the manuscript.

We thank the reviewer for the summary and the comments. Please find our comments below in return.

Major comments:

1 - The expression profile of PSMC3 in the eye, ear, and brain is not clear. The authors refer to ubiquitous expression, but high magnification images in studied tissues showing cellular expression (epithelial, neuronal, support, mesenchymal cells ...) are necessary.

These data are just verification data, which have been previously published by Thisse *et al* 2004 as part of an unbiased genome-wide screen of expression patterns (Thisse, B., Thisse, C. (2004), Fast Release Clones: A High Throughput Expression Analysis). These data were deposited in a publicly accessible database (Zebrafish Information Network (ZFIN), Direct Data Submission: <https://zfin.org/ZDB-PUB-040907-1>). Our *in situ* staining experiments confirmed the deposited data. This expression pattern is also consistent with that reported for the mouse orthologue in particular within the visual system, the auditory system and in the nervous system among other tissues (MGI source: 1098754). In human, the situation is very similar with a very large RNA and protein expression according to the human protein atlas (<https://www.proteinatlas.org/ENSG00000165916-PSMC3/>). For these reasons, we thus believe that a thorough assessment of PSMC3 expression would not bring any new insights in this regard. We hope the referee understands our choice and considers this concern as being addressed.

2- About zebrafish models, the authors describe the presence of 2 zebrafish psmc3 isoforms that share 83% sequence identity with the human orthologue. It was not clear if the MO used target one, or both isoforms.

This is indeed major information and we might have not been clear enough. The morpholino targets both psmc3 isoforms as indicated in the supplementary Figure S14A (splice morpholino).

Could the authors try a rescue using the human PSCM3 gene?

We appreciate the reviewer's comment; however rescuing the zebrafish KO using the human gene is only done when the zebrafish orthologous proteins cannot be obtained. Both zebrafish *psmc3* isoforms share 83% amino acid sequence identity with the human orthologue. To our point of view, it is not clear what could be learned from repeating this experiment with the human gene. Indeed, such cross-species experiments are extremely difficult to control. Different expression levels from the artificial expression systems, different codon usage etc or evolutionary adaptations in proteasome interactions may complicate the interpretation, thereby not telling us a lot about the function of the protein. Thus, in this study, we have performed the rescue with the wildtype zebrafish gene demonstrating specific phenotypes for our morpholino. Rescue was properly performed with a rescue rate of ~58% of the cases (n=60) for the cataract phenotype and 76% (n=60) for the ear phenotype (Figure 5B'+D'). For these reasons and we have followed the Editor's suggestion and rescued the patient's cells phenotype instead of the zebrafish's one.

3- The link between proteasome abnormalities observed in patients' cells and the morphological abnormalities observed in zebrafish is missing?

The point raised by the reviewer is valid and although we did not assess the proteasome activity here as in the patient's cells and concentrated on the morphological aspects in the zebrafish; we provide relevant information in the discussion to give more background on those aspects. See below:

"Interestingly, a reduction of proteasome activity has previously been associated with lens defects in zebrafish. The knock out of the zebrafish gene *psmd6* and the knock down of *psmd6* and *psmc2*, both encoding proteins of the proteasome, resulted in severe impairment of lens fibre development. Cataract was also noted as a consequence of disrupted lens fibre differentiation (Richardson *et al*, 2017). The ear phenotype of the *psmd6* mutant and both *psmd6* morphants were not assessed (Imai *et al.*, 2010). A direct link between the UPS and auditory hair cell death or impaired semicircular canal morphogenesis has not been described in zebrafish yet. However, knock down of *atoh1*, a gene regulated by the UPS, has been shown to severely affect hair cell development in the inner ear of zebrafish (Millimaki *et al.*, 2007). The malformations of canal pillars observed in zebrafish *psmc3* morphants and crispants might be also a secondary effect, as abnormal sensory cristae with few hair cells have been previously assumed to lead to an abnormal development of semicircular canals (Cruz *et al.*, 2009; Haddon and Lewis, 1991)."

4- Deafness is one major phenotype common to all 3 PSMC3-affected patients, It would be interesting to add a detailed phenotypic description (symmhair etry, age of onset, progression, & severity of the hearing loss) of the hearing loss. The fact that patients have cochlear implant may indicate profound hearing loss, but the presentation of patient audiogram (& age) could highlight some genotype-phenotype correlations that would be great to discuss and will add more depth to the paper. For instance, do the patients present (or have presented) signs of balance problems? Are there CT scans of MRI data from patients that support defects in circular canal formation? The origin of deafness is not clear. In the paper, the authors have extensively characterized the defects in semicircular canal morphogenesis, which has nothing to do with hearing.

We agree with the reviewer's comment and we provide additional information below. However, it is to notice that given the time of the initial analysis and the early cochlear implantation little data is available. For example, no audiogram could be retrieved before the cochlear implantation particularly because of the patients care outside of the hospitals (private medical doctors) and difficulty to establish full audiogram for very young children.

It is also remarkable that none of the family branches were connected to each other prior to their accidental meeting in the patients' waiting room in our reference center and after our examination and discovery of such rare combined conditions. This prevented any anticipation for the health care of the older children. All 3 patients had very early (in the first year of life) severe to profound bilateral hearing loss that required early cochlear implantation. Implantation was done for patients II.4 at 3 years and 3 months old, II.2 at 1 year and 10 months old and patient II.7 at 2 years and 4 months old.

In summary, OtoAcoustic Emissions (OAE) were positive at birth for each affected patients. However, deafness was suspected for all of them within the early months of life, respectively 8 months (II.4, II.2) and 1 year and 4 months (II.7). Auditory Brainstem Response (ABR) was in favor of profound deafness (no response at 110dB). MRI did not reveal any anomalies of the cochleovestibular nerves or labyrinthitis. However, temporal bone CT scan analysis of patient II.7 revealed lateral semicircular canal malformation (absence of the bony island in the right ear and small bony island in the left ear). It is hard to say whether the patients had any balance problem given the rest of the symptoms and if this could be linked to their neuropathy or the inner ear trouble. For sure, this was not obvious. Vestibular testing was impossible for patient II.4 and II.7. Patient II.2 had a preserved vestibular function: after cochlear implantation, vestibular testing showed presence and symmetrical responses for the lateral semicircular canals at middle (rotary-chair test) and low frequencies (caloric test).

In conclusion, deafness presented by the 3 affected children is either a progressive endocochlear deafness or an auditory neuropathy (cochlear response but no signal transmission) or more likely a combination of both. The manuscript and Figure 1 have been modified to reflect this data. We hope that we have clarified the deafness aspects for the reviewer.

Figure legend: Temporal bone CT-scan from patient II.7 (left column) and a normal scan (right column). The left ear is shown on the upper panels while the right ear on the lower panels

More information on the morphology and function of the hair cells would be more informative. Besides showing reduction of kinocilia, what about hair cells (& support cells) morphology? hair bundle architecture? The authors could use FM1-43 that provide indication of hair cells activity.

Following the reviewer's suggestion, we stained 5 dpf wild-type and morphant zebrafish embryos with FM1-43 and observed hair cells. The hair cell morphology did not appear affected in morphants. However, we observed a decrease of hair cell length in morphant embryos, which may contribute to ear deficiency.

(A) Hair cell and stereo/kinocilia of 5 dpf wild-type (wt) and morphants (mo) embryos after incubation in FM1-43.

(B) Graph showing cilia length in μm . P-value: 0.022.

(C) Shape of hair cell body. No differences are noted.

In line with these results, we added and modified the following paragraph in the manuscript: "The inner ear possesses hair cells to sense both vestibular and auditory stimuli. These apical structures consist of a bundle of villi-like structures called stereocilia and kinocilia, collectively referred to as a hair bundle. Because these cilia have been shown to play a key role at least in mechanosensation during development (Kindt et al., 2012), we immunostained crispants (sgRNA2 injected with Cas9) and control injected embryos (sgRNA2 without Cas9) at 5 dpf using anti-acetylated tubulin antibody. In 40% of crispants (n=15), a reduced number of cilia was observed while their number in control injected embryos (n=10) was similar to that of uninjected embryos (n=10) (Figure 6E+E'). In order to examine the morphology of hair cells themselves, we used FM1-43. This did not reveal any obvious difference between wild-type and morphant embryos. However, we observed that the length of the cilia was decreased when compared to wild-type embryos (data not shown)."

Minor comments:

2- Patients present some variability in phenotype; what's the authors explanation for those differences?

We thank the reviewer for pointing this out. This is always very intriguing to observe subtle or major phenotypic variability while having the same gene mutated and in the present case even the same

pathogenic variation. However, this is not an isolated case. For instance, having worked in the ciliopathy field for many years, especially on the Bardet-Biedl syndrome (BBS; MIM 209900), there is a single pathogenic variation (founder European effect, c.1169T>G, p.M390R) in *BBS1* that can lead to the full spectrum of the disease (*retinitis pigmentosa*, postaxial polydactyly, obesity, hypogonadism, cognitive impairment and kidney dysfunction) or “only” an isolated *retinitis pigmentosa*. Even in the same family with multiple affected siblings, there is variability in their phenotype.

The observed variability can be linked to the genetic background of each individual. In this large family of consanguineous origin only a few shared homozygous regions were observed. In fact, the region of 2.12 Mb encompassing *PSMC3* on chromosome 11 is the largest observed (Figure S1). In line with that, one cannot exclude modifier genes that would enhance or moderate some phenotypic aspects although nothing was obvious could be observed and would require more families and individuals to be explored. Among the other possible explanations, differences in their respective gene expression patterns or their respective environment can possibly influence their conditions.

4- What about cell death (TUNEL, caspase ...) in the eye and ear of zebrafish models?

The number of apoptotic cells detected with the TUNEL assay in the lens of morphants appears to be slightly smaller than that in wild-type embryos at 3 dpf. However these results are not significant (see p-value). The same tendency is observed in the ear. In conclusion, cataract or deafness cannot be explained by an increased apoptosis.

Number of TUNEL-positive cells in lenses. N=9 for each conditions.

p-value for lens: 0.52

p-value for ears: 0.32

In line with these results, we added the following sentence to the manuscript: “The observed cataract was not due to increased apoptosis, as TUNEL staining did not reveal more positive nuclei in the morphant compared to wild-type (data not shown). “

3- It's not clear if haploinsufficiency is the sole mechanism taking place in patients. Heterozygotes (50% of protein) are normal. Patients cells show no difference in protein localization, but it's not clear how much wild-type protein (or transcript) is present despite the PSMC3 pathogenic variant. Is the truncated protein still present or not? If the truncated form might have some gain (or semi-dominant) function is not clear.

We agree with the reviewer as we do not have a definitive answer on this but only hypothesis. Further families and patients with additional pathogenic alleles would be necessary for sorting this out. We have tried to clarify this in the updated version of the manuscript. Situation is complex given the effect on the proteasome regulation.

As we have pointed out in the discussion, *PSMC3* is predicted to be extremely intolerant to loss of function (LoF) variations as the other reported proteasome genes causing diseases. In other words, only *de novo* LoF would be expected for such gene.

The effect of the homozygous variation is to incorporate a novel cryptic exon leading to a frameshift in the coding sequence of the transcript. There are three facts that indicate an additional (semi-)dominant negative effect (although only minor) of the altered version missing the C-terminus of the protein, which is important for proteasome assembly and function:

1. We observe minor expression of the transcript with the cryptic exon (Figure 1E).
2. In Figure 3C (native PAGE analysis) there is a clear accumulation of 19S precursor complexes in the patient's cells indicating assembly problems (lower bands).
3. In Figure 3D in the immunoblots for Rpt5 there is faint band coming up in the patient's sample only, which runs below of the correct size of Rpt5 (i.e. 49,203.54 Da) and corresponds to the truncated form in its size ((i.e. 43,458.95 Da using longer exposure time).

It is well known that haploinsufficiency describes a single allele (by extension 50% of the protein) not sufficient to maintain its function in a given cellular process. In our situation, we hypothesize that each allele is maybe contributing only to 25% of damaging allele which when heterozygous does not lead to defects in the cell and thus no phenotype. However at the homozygous state, we have 50% (combined) of damaging allele that is enough to cause the phenotype. This would also fit with a so called

hypomorphic allele. The difficulty to analyze such cases has been reported very recently (Misra *et al*, 2020; Monies *et al*, 2017).

1- There are many typographical errors (missing spaces; most references positioned after the "."; ref TANAKA (not in capitals),...)

e.g. Page 9: last line:

... in the inner ear of zebrafish.(Millimaki et al., 2007) The of canal pillars observed in zebrafish psmc3

We thank the reviewer for pointing this out. We have used the Zotero system to integrate and format the references in the manuscript and it seems to have some issues. We have now corrected this and we hope that none have been missed.

Referee #2 (Comments on Novelty/Model System for Author):

This is an interesting paper. If the authors follow my (easy) suggestions the technical quality would move to high. This is a new example of another proteasomal subunit that -when mutated- would cause neurological pathologies.

Referee #2 (Remarks for Author):

This is an interesting manuscript where the authors identified a mutation in the 19S proteasomal subunit PSMC3 that relates to malformation and a range of neurological disorders. They report that this mutation results in an accumulation of ubiquitinated species in patient fibroblasts, which is -as expected- not associated to the activity of the proteasome but likely the transfer into the proteasomal proteolytic cage for degradation. As a result, the patient fibroblasts do not really respond to proteotoxic stress. Importantly, the authors could mimic the patient's phenotypes in zebrafish where the PSMC3 was eliminated by morpholinos.

This is important work and reveals another proteasome related case for neurological diseases. I have only a few points.

We thank the reviewer for the kind words and for the comments. Please find our comments in return.

1. The data are shown with fibroblasts from only one (out of three) patients and it would be better to show at least one other patient's fibroblast as well.

We appreciate the reviewer's comment and we also wished we had access to further samples of affected patients. Unfortunately, we had one ethical agreement for taking a skin biopsy for only patient II.4, while the others branches did not want to do so. Nevertheless, we have tried again after the reviewer's comment and they did not change their mind.

2. The experiments would be even better when normal psmc3 is overexpressed in the fibroblast to compete mutant psmc3 away and then show that the phenotypes (ubiquitin accumulation, responses to proteolytic stress) are corrected.

We have followed the referee's excellent suggestion and performed a series of rescue experiments in which patient fibroblasts were transfected with constructs expressing wild-type PSMC3 driven by CMV promoter. These results are now integrated in a novel Figure 5. Indeed, PSMC3/Rpt5 is highly expressed upon transfection. Our data show that PSMC3/Rpt5 overexpression resulted in decreased accumulation of K48-linked ubiquitin-protein conjugates in mutant cells, as determined by western-blotting (Figure 5A). Densitometry analysis of four independent experiments revealed that the intracellular concentration of ubiquitin-modified proteins was reduced by about 20% in patient cells following PSMC3 transfection (Figure 5B), thereby confirming the decisive role of the *PSMC3* homozygous mutation in perturbing protein homeostasis. In addition, restoring wild-type PSMC3 in PSMC3 mutant fibroblasts led to a decreased overload of the TCF11/Nrf1 pathway, as evidenced by decreased processing/activation of the TCF11/Nrf1 transcription factor under these conditions (Figure 5C). Most importantly, this was further accompanied by the capacity of the patient cells to rescue proteasome subunit expression in response to proteotoxic stress initiated by carfilzomib (Figure 5C). Altogether, these data clearly identify the deep intronic homozygous *PSMC3* variation as the genetic cause for the failure of the cells to preserve protein homeostasis under proteotoxic stress. This point is now addressed in the revised version of the manuscript (Figure 5, page 6, lines 237-250 and page 9, lines 385-390).

3. PA28 is highly expressed in the patient fibroblasts. What is the reason for this effect? And why does it decrease following proteotoxic stress?

The reviewer raises a valid point here, as the PA28 expression profile observed in patient's cells clearly differs from that detected in control fibroblasts. Noteworthy and apart from PA28, patient fibroblasts carrying the deep intronic homozygous *PSMC3* variation are also endowed with higher amounts of immunoproteasome subunits (i.e. $\beta 5i$ and $\beta 1i$), as determined by western blotting (Figure 4). The upregulation of PA28 and immunoproteasome subunits in these cells might reflect a mechanism destined to compensate the inefficiency of mutant proteasomes at eliminating damaged proteins. It is indeed understood that 26S immunoproteasomes are more effective than their standard counterparts in clearing ubiquitin-marked proteins (Seifert *et al*, 2010; Ebstein *et al*, 2013; St-Pierre *et al*, 2017). Likewise, it has been shown that PA28-20S complexes are more efficient than free 20S proteasomes at removing oxidant-damaged proteins (Pickering *et al*, 2010; Li *et al*, 2010). It is therefore seductively easy to imagine that patient cells upregulate PA28 and immunoproteasomes in order to assist their mutant proteasomes to cope with protein aggregates. The process leading to the upregulation of both PA28 and immunoproteasome subunits in patient fibroblasts remains unclear but may conceivably rely on a type I interferon (IFN) autocrine loop which is frequently detected in cells suffering from proteasome loss-of-function mutations (Brehm *et al*, 2015; Poli *et al*, 2018). The reason why the steady-state expression level of PA28- α drops following proteasome inhibition in patient fibroblasts is not addressed in this manuscript but may be explained by the overall decrease of proteasome subunits observed in these cells under these conditions. Because PA28 physically associates with 20S core particles to form PA28-20 proteasome complexes, it is highly likely that PA28- α undergoes degradation together with the 20S proteasome subunits. These points are now clarified in the revised version of the manuscript (page 6, lines 227-229 and 238-239 as well as page 9, lines 405-407).

In this context it is also interesting to note that type I interferon impacts stem cell function (Eggenberger *et al*, 2019; Yu *et al*, 2015).

References

- Brehm A, Liu Y, Sheikh A, Marrero B, Omoyinmi E, Zhou Q, Montealegre G, Biancotto A, Reinhardt A, Almeida de Jesus A, Pelletier M, Tsai WL, Remmers EF, Kardava L, Hill S, Kim H, Lachmann HJ, Megarbane A, Chae JJ, Brady J, et al (2015) Additive loss-of-function proteasome subunit mutations in CANDLE/PRAAS patients promote type I IFN production. *The Journal of clinical investigation* **125**: 4196–211
- Ebstein F, Voigt A, Lange N, Warnatsch A, Schröter F, Prozorovski T, Kuckelkorn U, Aktas O, Seifert U, Kloetzel P-M & Krüger E (2013) Immunoproteasomes Are Important for Proteostasis in Immune Responses. *Cell* **152**: 935–937
- Eggenberger J, Blanco-Melo D, Panis M, Brennand KJ & tenOever BR (2019) Type I interferon response impairs differentiation potential of pluripotent stem cells. *Proc Natl Acad Sci USA* **116**: 1384
- Li J, Powell SR & Wang X (2010) Enhancement of proteasome function by PA28 α overexpression protects against oxidative stress. *The FASEB Journal* **25**: 883–893
- Misra S, Hassanali N, Bennett AJ, Juszczak A, Caswell R, Colclough K, Valabhji J, Ellard S, Oliver NS & Gloyn AL (2020) Homozygous Hypomorphic *HNF1A* Alleles Are a Novel Cause of Young-Onset Diabetes and Result in Sulfonylurea-Sensitive Diabetes. *Diabetes Care* **43**: 909
- Monies D, Maddirevula S, Kurdi W, Alanazy MH, Alkhalidi H, Al-Owain M, Sulaiman RA, Faqeih E, Goljan E, Ibrahim N, Abdulwahab F, Hashem M, Abouelhoda M, Shaheen R, Arold ST & Alkuraya FS (2017) Autozygosity reveals recessive mutations and novel mechanisms in dominant genes: implications in variant interpretation. *Genetics in Medicine* **19**: 1144–1150
- Pickering AM, Koop AL, Teoh CY, Ermak G, Grune T & Davies KJA (2010) The immunoproteasome, the 20S proteasome and the PA28 $\alpha\beta$ proteasome regulator are oxidative-stress-adaptive proteolytic complexes. *Biochemical Journal* **432**: 585–595

- Poli MC, Ebstein F, Nicholas SK, de Guzman MM, Forbes LR, Chinn IK, Mace EM, Vogel TP, Carisey AF, Benavides F, Coban-Akdemir ZH, Gibbs RA, Jhangiani SN, Muzny DM, Carvalho CMB, Schady DA, Jain M, Rosenfeld JA, Emrick L, Lewis RA, et al (2018) Heterozygous Truncating Variants in POMP Escape Nonsense-Mediated Decay and Cause a Unique Immune Dysregulatory Syndrome. *American journal of human genetics* **102**: 1126–1142
- Richardson R, Tracey-White D, Webster A & Moosajee M (2017) The zebrafish eye—a paradigm for investigating human ocular genetics. *Eye* **31**: 68–86
- Seifert U, Bialy LP, Ebstein F, Bech-Otschir D, Voigt A, Schröter F, Prozorovski T, Lange N, Steffen J, Rieger M, Kuckelkorn U, Aktas O, Kloetzel P-M & Krüger E (2010) Immunoproteasomes Preserve Protein Homeostasis upon Interferon-Induced Oxidative Stress. *Cell* **142**: 613–624
- St-Pierre C, Morgand E, Benhammadi M, Rouette A, Hardy M-P, Gaboury L & Perreault C (2017) Immunoproteasomes Control the Homeostasis of Medullary Thymic Epithelial Cells by Alleviating Proteotoxic Stress. *Cell Reports* **21**: 2558–2570
- Yu Q, Katlinskaya YV, Carbone CJ, Zhao B, Katlinski KV, Zheng H, Guha M, Li N, Chen Q, Yang T, Lengner CJ, Greenberg RA, Johnson FB & Fuchs SY (2015) DNA-Damage-Induced Type I Interferon Promotes Senescence and Inhibits Stem Cell Function. *Cell Reports* **11**: 785–797

20th Apr 2020

Dear Dr. Muller,

Thank you for the submission of your revised manuscript to EMBO Molecular Medicine. We have now received the enclosed report from the referee who was asked to re-assess it. As you will see this reviewer is now supportive and I am pleased to inform you that we will be able to accept your manuscript pending the following final amendments:

1) Please address the minor comments of referee 1 and expand the discussion.

Please provide a point-by-point letter INCLUDING my comments as well as the reviewer's reports and your detailed responses to their comments (as Word file).

2) Please carefully check the authors guidelines for formatting your supplemental information:

Expanded view and Appendix (see:

<https://www.embopress.org/page/journal/17574684/authorguide#expandedview>)

-We need a Table of Content as the 1st page of the Appendix and the nomenclature should be corrected to "Appendix Figure S1" etc. and "Appendix Table S1" etc.

-For the movie, legend should be removed from appendix and zipped together with the movie file.

Nomenclature should be corrected to "Movie EV1"

-appendix figures are of rather poor quality, please try to improve the resolution of the figures.

-fig s10D is missing magnification scale bar

3) Source Data:

We encourage the publication of source data, particularly for electrophoretic gels, blots, but also microscopy images with the aim of making primary data more accessible and transparent to the reader. Would you be willing to provide a PDF file per figure that contains the original, uncropped and unprocessed scans of all or key gels used in the figure (including molecular weight markers)? The PDF files should be labeled with the appropriate figure/panel number (1 file/figure), and should have molecular weight markers; further annotation may be useful but is not essential. The PDF files will be published online with the article as supplementary "Source Data" files. If you have any questions regarding this just contact me.

4) In the main manuscript file, please do the following:

- correct/answer the track changes suggested by our data editors by working from the attached document

- add up to 5 keywords

- in M&M, a statistical paragraph is needed and it should reflect all information that you have filled in the Authors checklist, especially regarding randomisation, blinding, replication.

- indicate in legends exact $n=$ and exact $p=$ values, not a range, along with the statistical test used. Some people found that to keep the figures clear, providing an Appendix table Sx with all exact p -values was preferable. You are welcome to do this if you want to.

- revise the cal out for figure 4A to figure 4

- move Table 1 before figure legends

- spell out "ENT" line 488

- remove "data not shown" line 284. As per our guidelines, on "Unpublished Data" the journal does

not permit citation of "Data not shown". All data referred to in the paper should be displayed in the main or Expanded View figures. "Unpublished observations" may be referred to in exceptional cases, where these are data peripheral to the major message of the paper and are intended to form part of a future or separate study, the names of the persons that reported the observation should be listed in brackets. Personal communications (Author name(s), personal communications) must be authorised in writing by those involved, and the authorisation sent to the editorial office at time of submission.

5) Funding:

Please make sure to indicate in our submission system all sources of funding including grant numbers and to whom they are allocated.

6) Authors' contribution: the contribution of every author must be detailed in a separate section. Make sure to differentiate the contribution of Fouzia Studer and Florian Sandron

7) The Paper Explained should be included in the main article, after For More Information. Please label the subsections as Problem, Results, Impact

8) Synopsis.

I have slightly modified the text, would the following be of for you?

Synopsis

Whole genome sequencing in a large consanguineous family with neurosensory syndrome including revealed a unique homozygous deep intronic pathogenic variant in PSMC3, encoding one of the proteasome subunit. Further in vitro and in vivo analyses confirmed the pathogenicity of the PSMC3 mutation.

Bullet points

- This is the first implication of a 26S proteasome AAA-ATPase of the 19S proteasome regulatory complex in a neurosensory disease with early onset cataract and deafness.
- Functional analysis using patient's cells revealed a pathogenic mechanism with proteasome impairment resulting in proteotoxic stress with over-activation of the TCF11/Nrf1 transcriptional pathway.
- Zebrafish model reproduces the human phenotype with cataract and ear malformations.
- PSMC3 plays a major role in inner ear, lens and central nervous system development
- These results expand our knowledge on the genetic background of the emerging proteasomopathy.

Regarding the synopsis image, would it be possible to increase the font in the image provided. As it is, it will be hardly readable when resized for the website.

9) As part of the EMBO Publications transparent editorial process initiative (see our Editorial at <http://embomolmed.embopress.org/content/2/9/329>), EMBO Molecular Medicine will publish online a Review Process File (RPF) to accompany accepted manuscripts.

In the event of acceptance, this file will be published in conjunction with your paper and will include the anonymous referee reports, your point-by-point response and all pertinent correspondence relating to the manuscript. Let us know whether you agree with the publication of the RPF and as

here, if you want to remove or not any figures from it prior to publication.

10) Please note that we now mandate that all corresponding authors list an ORCID digital identifier. This takes less than 90 seconds to complete. We encourage all authors to supply an ORCID identifier, which will be linked to their name for unambiguous name identification.

11) Data availability

Please relabel the section and format as following:

- [data type]: [full name of the resource] [accession number/identifier] ([doi or URL or identifiers.org/DATABASE:ACCESSION])

examples:

* RNA-Seq data: Gene Expression Omnibus GSExxxxx

(<https://www.ncbi.nlm.nih.gov/geo/query/acc.cgi?acc=GSExxxxx>)

* Chip-Seq data: Gene Expression Omnibus GSEyyyyy

(<https://www.ncbi.nlm.nih.gov/geo/query/acc.cgi?acc=GSEyyyyy>)

* patients' sequences: Database of Genotypes and Phenotypes (dbGAP) Xxxxxx

(https://www.ncbi.nlm.nih.gov/projects/gap/cgi-bin/study.cgi?study_id=Xxxxxx)

* Protein interaction AP-MS data: PRIDE PXD000xxx

(<http://www.ebi.ac.uk/pride/archive/projects/PXD000xxx>)

* Imaging dataset: Image Data Resource doi:10.17867/10000xxx (<http://doi.org/10.17867/10000xxx>)

Please submit your revised manuscript within two weeks. I look forward to seeing a revised form of your manuscript as soon as possible.

I look forward to reading a new revised version of your manuscript as soon as possible.

Yours sincerely,

Celine Carret

Celine Carret, PhD

Senior Editor

EMBO Molecular Medicine

*** Instructions to submit your revised manuscript ***

In the event of acceptance, this file will be published in conjunction with your paper and will include

the anonymous referee reports, your point-by-point response and all pertinent correspondence relating to the manuscript. If you do NOT want this file to be published, please inform the editorial office at contact@embomolmed.org.

To submit your manuscript, please follow this link:

Link Not Available

- 1) a .doc formatted version of the manuscript text (including Figure legends and tables)
- 2) Separate figure files*
- 3) supplemental information as Expanded View and/or Appendix. Please carefully check the authors guidelines for formatting Expanded view and Appendix figures and tables at <https://www.embopress.org/page/journal/17574684/authorguide#expandedview>
- 4) a letter INCLUDING the reviewer's reports and your detailed responses to their comments (as Word file).
- 5) Author contributions: the contribution of every author must be detailed in a separate section.
- 6) Every published paper now includes a 'Synopsis' to further enhance discoverability. Synopses are displayed on the journal webpage and are freely accessible to all readers. They include a short stand first (maximum of 300 characters, including space) as well as 2-5 one sentence bullet points that summarise the paper. Please write the bullet points to summarise the key NEW findings. They should be designed to be complementary to the abstract - i.e. not repeat the same text. We encourage inclusion of key acronyms and quantitative information (maximum of 30 words / bullet point). Please use the passive voice. Please attach these in a separate file or send them by email, we will incorporate them accordingly.

You are also welcome to suggest a striking image or visual abstract to illustrate your article. If you do please provide a jpeg file 550 px-wide x 400-px high.

7) The system will prompt you to fill in your funding and payment information. This will allow Wiley to send you a quote for the article processing charge (APC) in case of acceptance. This quote takes into account any reduction or fee waivers that you may be eligible for. Authors do not need to pay any fees before their manuscript is accepted and transferred to our publisher.

Photos 400-800 DPI

*Additional important information regarding figures and illustrations can be found at <http://bit.ly/EMBOPressFigurePreparationGuideline>

The system will prompt you to fill in your funding and payment information. This will allow Wiley to send you a quote for the article processing charge (APC) in case of acceptance. This quote takes into account any reduction or fee waivers that you may be eligible for. Authors do not need to pay any fees before their manuscript is accepted and transferred to our publisher.

***** Reviewer's comments *****

Referee #1 (Remarks for Author):

The revised manuscript is much improved and the authors have responded quiet appropriately to most raised issues.

The precise causal mechanisms of some reported patient phenotypes still warrant further studies, but the authors provide a substantial amount of new data that deserve publication in the EMM journal.

Minor points:

The discussion of inner ear related data can be further improved.

Based on new data, patients otoacoustic emissions were positive at birth, which indicate that the auditory outer hair cells are functional, despite possible kinocilium defects that might have occurred based on zebrafish data.

Positive otoacoustic emissions with abnormal Auditory brainstem responses are hallmarks of auditory neuropathies. Most of auditory neuropathies are not eligible to cochlear implantation (CI). The authors might discuss CI outcome performances in the PSMC3 patients. This might provide information as to the impact of PSMC3 beyond the peripheral inner ear?

20th Apr 2020

Dear Dr. Muller,

Thank you for the submission of your revised manuscript to EMBO Molecular Medicine. We have now received the enclosed report from the referee who was asked to re-assess it. As you will see this reviewer is now supportive and I am pleased to inform you that we will be able to accept your manuscript pending the following final amendments:

Dear editor, this is very good news and we hope that we have now replied to all of the points. See our comments (in red) below.

1) Please address the minor comments of referee 1 and expand the discussion.

This has been done. See our replies in the corresponding section.

Please provide a point-by-point letter INCLUDING my comments as well as the reviewer's reports and your detailed responses to their comments (as Word file).

2) Please carefully check the authors guidelines for formatting your supplemental information: Expanded view and Appendix (see: <https://www.embopress.org/page/journal/17574684/authorguide#expandedview>)

-We need a Table of Content as the 1st page of the Appendix and the nomenclature should be corrected to "Appendix Figure S1" etc. and "Appendix Table S1" etc.

Ok this has been done.

-For the movie, legend should be removed from appendix and zipped together with the movie file. Nomenclature should be corrected to "Movie EV1"

Ok this has been done.

-appendix figures are of rather poor quality, please try to improve the resolution of the figures.

We are surprised by this and it must be linked to some default Word document settings (automatic image compression) that has been changed now. Figures have also been regenerated with a higher resolution and incorporated again.

-fig s10D is missing magnification scale bar

Ok this has been added and corrected in the corresponding figure legend.

3) Source Data:

We encourage the publication of source data, particularly for electrophoretic gels, blots, but also microscopy images with the aim of making primary data more accessible and transparent to the reader. Would you be willing to provide a PDF file per figure that contains the original, uncropped and unprocessed scans of all or key gels used in the figure (including molecular weight markers)? The PDF files should be labeled with the appropriate figure/panel number (1 file/figure), and should have molecular weight markers; further annotation may be useful but is not essential. The PDF files will be published online with the article as supplementary "Source Data" files. If you have any questions regarding this just contact me.

We fully agree on this point. This is a very important part of science and we have now assembled all source data that were required.

4) In the main manuscript file, please do the following:

- correct/answer the track changes suggested by our data editors by working from the attached document

Ok this has been done.

- add up to 5 keywords

Ok this has been done.

- in M&M, a statistical paragraph is needed and it should reflect all information that you have filled in the Authors checklist, especially regarding randomisation, blinding, replication.

Ok this has been done and the paragraph is now the last part of this section.

- indicate in legends exact n= and exact p= values, not a range, along with the statistical test used. Some people found that to keep the figures clear, providing an Appendix table Sx with all exact p-values was preferable. You are welcome to do this if you want to.

Ok this has been done in the manuscript and in the Appendix Data. We provide now an Appendix Table S8 including the exact p-values when not available already in the figures.

- revise the call out for figure 4A to figure 4

Ok this has been done.

- move Table 1 before figure legends

Ok this has been done.

- spell out "ENT" line 488

Ok ENT mean "Ear-Nose-Throat". This has been spelled out in the updated version of the manuscript.

- remove "data not shown" line 284. As per our guidelines, on "Unpublished Data" the journal does not permit citation of "Data not shown". All data referred to in the paper should be displayed in the main or Expanded View figures. "Unpublished observations" may be referred to in exceptional cases, where these are data peripheral to the major message of the paper and are intended to form part of a future or separate study, the names of the persons that reported the observation should be listed in brackets. Personal communications (Author name(s), personal communications) must be authorised in writing by those involved, and the authorisation sent to the editorial office at time of submission.

We understand this. The "data not shown" was referring to the additional data provided during the reviewing process. This has been now included as Appendix Figure S10 and S15.

5) Funding:

Please make sure to indicate in our submission system all sources of funding including grant numbers and to whom they are allocated.

We will take make sure all will be done properly.

6) Authors' contribution: the contribution of every author must be detailed in a separate section. Make sure to differentiate the contribution of Fouzia Studer and Florian Sandron

Ok this has been done. Fouzia Studer is abbreviated as F.S. while Florian Sandron as F.Sa.

7) The Paper Explained should be emoted in the main article, after For More Information. Please label the subsections as Problem, Results, Impact

Ok this has been done.

8) Synopsis.

I have slightly modified the text, would the following be of for you?

This is ok for us. We would simply remove the word "including" highlighted in red below as it does not fit anymore. We have now removed the file from the online data deposit as it is included in this file in an updated version.

Synopsis

Whole genome sequencing in a large consanguineous family with neurosensory syndrome **including** revealed a unique homozygous deep intronic pathogenic variant in PSMC3, encoding one of the proteasome subunit. Further in vitro and in vivo analyses confirmed the pathogenicity of the PSMC3 mutation.

Bullet points

- This is the first implication of a 26S proteasome AAA-ATPase of the 19S proteasome regulatory complex in a neurosensorial disease with early onset cataract and deafness.
- Functional analysis using patient's cells revealed a pathogenic mechanism with proteasome impairment resulting in proteotoxic stress with over-activation of the TCF11/Nrf1 transcriptional pathway.
- Zebrafish model reproduces the human phenotype with cataract and ear malformations.
- PSMC3 plays a major role in inner ear, lens and central nervous system development
- These results expand our knowledge on the genetic background of the emerging proteasomopathy.

Regarding the synopsis image, would it be possible to increase the font in the image provided. As it is, it will be hardly readable when resized for the website.

Ok this has been done in the second version provided. Annotations have been simplified to increase the font. Please let us know if it still needs some improvements.

9) As part of the EMBO Publications transparent editorial process initiative (see our Editorial at <http://embomolmed.embopress.org/content/2/9/329>), EMBO Molecular Medicine will publish online a Review Process File (RPF) to accompany accepted manuscripts.

In the event of acceptance, this file will be published in conjunction with your paper and will include the anonymous referee reports, your point-by-point response and all pertinent correspondence relating to the manuscript. Let us know whether you agree with the publication of the RPF and as here, if you want to remove or not any figures from it prior to publication.

Ok this is perfect, we agree on the transparency of the reviewing.

10) Please note that we now mandate that all corresponding authors list an ORCID digital identifier. This takes less than 90 seconds to complete. We encourage all authors to supply an ORCID identifier, which will be linked to their name for unambiguous name identification.

Ok this should have been done already by all corresponding authors. However, you can find below the list those if not found automatically:

Elke Krüger 0000-0002-2551-242X
Jean Muller 0000-0002-7682-559X
Uwe Strähle 0000-0002-4062-9431
Hélène Dollfus 0000-0002-2249-895X

11) Data availability

Please relabel the section and format as following:

- [data type]: [full name of the resource] [accession number/identifier] ([doi or URL or identifiers.org/DATABASE:ACCESSION])

examples:

* RNA-Seq data: Gene Expression Omnibus GSExxxxx
(<https://www.ncbi.nlm.nih.gov/geo/query/acc.cgi?acc=GSExxxxx>)

* Chip-Seq data: Gene Expression Omnibus GSEyyyyy
(<https://www.ncbi.nlm.nih.gov/geo/query/acc.cgi?acc=GSEyyyyy>)

* patients' sequences: Database of Genotypes and Phenotypes (dbGAP) Xxxxxx
(https://www.ncbi.nlm.nih.gov/projects/gap/cgi-bin/study.cgi?study_id=Xxxxxx)

* Protein interaction AP-MS data: PRIDE PXD000xxx
(<http://www.ebi.ac.uk/pride/archive/projects/PXD000xxx>)

* Imaging dataset: Image Data Resource doi:10.17867/10000xxx (<http://doi.org/10.17867/10000xxx>)

Ok this has been done. All datasets will be made or are already public. The ClinVar data will be released upon publication and availability in PubMed.

Please submit your revised manuscript within two weeks. I look forward to seeing a revised form of your manuscript as soon as possible.

I look forward to reading a new revised version of your manuscript as soon as possible.

Yours sincerely,

Celine Carret

Celine Carret, PhD
Senior Editor
EMBO Molecular Medicine

***** Reviewer's comments *****

Referee #1 (Remarks for Author):

The revised manuscript is much improved and the authors have responded quite appropriately to most raised issues.

The precise causal mechanisms of some reported patient phenotypes still warrant further studies, but the authors provide a substantial amount of new data that deserve publication in the EMM journal.

We thank the reviewer for the kind words. Please find below our additional comments in return.

Minor points:

The discussion of inner ear related data can be further improved.

Based on new data, patients otoacoustic emissions were positive at birth, which indicate that the auditory outer hair cells are functional, despite possible kinocilium defects that might have occurred based on zebrafish data.

Positive otoacoustic emissions with abnormal Auditory brainstem responses are hallmarks of auditory neuropathies. Most of auditory neuropathies are not eligible to cochlear implantation (CI). The authors might discuss CI outcome performances in the PSMC3 patients. This might provide information as to the impact of PSMC3 beyond the peripheral inner ear?

Indeed, OtoAcoustic Emissions (OAE) were present at birth, indicating that outer hair cells were initially functional. But at the time deafness was diagnosed (8 month and 1 year and 3 month old), otoacoustic emissions were no longer recorded: transitory-evoked OAE were not present for patient II.2 and II.7 and Distortion-Product OAE (DP-OAE) were not present for patient II.4 who underwent full auditory examination under general anesthesia in Belgium in July 2005. During this examination, a neuropathic component was evoked after recording a cochlear microphonic on both sides (during Auditory Brainstem Response, using separate runs of condensation and rarefaction polarity clicks), even though DP-OAE were absent (with the assumption that the DP-OAE were absent due to the presence of grommets). However, lack of additional information did not help to either confirm or infirm this hypothesis and discordance between tone and vocal audiometry cannot be established due to autistic features and severe language delay.

According to your remarks, we have now exposed the cochlear implant outcome performances in the discussion.

7th May 2020

Dear Dr. Muller,

We are pleased to inform you that your manuscript is accepted for publication and will be soon sent to our publisher to be included in the next available issue of EMBO Molecular Medicine.

Before we proceed however, could you please clarify the following: in the figure 3, the molecular weight for PSMC1 is indicated at ~ 40kDa but it is higher in the source data file. Please amend the figure 3 accordingly and send the new one to us by email.

Can you please also encourage your co-corresponding author Dr. Strähle to add his ORCID number? We won't be able to move forward without these. Thank you for your cooperation and understanding.

Please read below for additional IMPORTANT information regarding your article, its publication and the production process.

Congratulations on your interesting work,

Celine Carret

Celine Carret, PhD
Senior Editor
EMBO Molecular Medicine

Follow us on Twitter @EmboMolMed
Sign up for eTOCs at embopress.org/alertsfeeds

*** ** IMPORTANT INFORMATION ** **

SPEED OF PUBLICATION

The journal aims for rapid publication of papers, using using the advance online publication "Early View" to expedite the process: A properly copy-edited and formatted version will be published as "Early View" after the proofs have been corrected. Please help the Editors and publisher avoid delays by providing e-mail address(es), telephone and fax numbers at which author(s) can be contacted.

Should you be planning a Press Release on your article, please get in contact with embomolmed@wiley.com as early as possible, in order to coordinate publication and release dates.

LICENSE AND PAYMENT:

All articles published in EMBO Molecular Medicine are fully open access: immediately and freely available to read, download and share.

EMBO Molecular Medicine charges an article processing charge (APC) to cover the publication costs. You, as the corresponding author for this manuscript, should have already received a quote with the article processing fee separately. Please let us know in case this quote has not been received.

Once your article is at Wiley for editorial production you will receive an email from Wiley's Author Services system, which will ask you to log in and will present you with the publication license form for completion. Within the same system the publication fee can be paid by credit card, an invoice, pro forma invoice or purchase order can be requested.

Payment of the publication charge and the signed Open Access Agreement form must be received before the article can be published online.

PROOFS

You will receive the proofs by e-mail approximately 2 weeks after all relevant files have been sent to our Production Office. Please return them within 48 hours and if there should be any problems, please contact the production office at embopressproduction@wiley.com.

Please inform us if there is likely to be any difficulty in reaching you at the above address at that time. Failure to meet our deadlines may result in a delay of publication.

All further communications concerning your paper proofs should quote reference number EMM-2019-11861-V3 and be directed to the production office at embopressproduction@wiley.com.

Thank you,

Celine Carret, PhD
Senior Editor
EMBO Molecular Medicine

Corresponding Author Name: Elke Krüger Jean Müller, Uwe Strähle, Hélène Dollfus

Manuscript Number: EMM-2019-11861